# EvolProver: Advancing Automated theorem proving by Evolving Formalized Problems via Symmetry and Difficulty

**Yuchen Tian**[1,2][*][†]   **Ruiyuan Huang**[3,2][*][‡]  **Xuanwu Wang**[1]    **Jing Ma**[1][§]
**Zengfeng Huang**[3,4]    **Ziyang Luo**[1]    **Hongzhan Lin**[1]    **Da Zheng**[2][§]  **Lun Du**[2][§]
[1]Hong Kong Baptist University
[2]Ant Group
[3]School of Data Science, Fudan University
[4]Shanghai Innovation Institute

## Abstract

Large Language Models (LLMs) for formal theorem proving have shown significant promise, yet they often lack generalizability and are fragile to even minor transformations of problem statements. To address this limitation, we introduce a novel data augmentation pipeline designed to enhance model robustness from two perspectives: symmetry and difficulty. From the symmetry perspective, we propose two complementary methods: **EvolAST**, an Abstract Syntax Tree (AST) based approach that targets syntactic symmetry to generate semantically equivalent problem variants, and **EvolDomain**, which leverages LLMs to address semantic symmetry by translating theorems across mathematical domains. From the difficulty perspective, we propose **EvolDifficulty**, which uses carefully designed evolution-based instructions to guide LLMs in generating new theorems with a wider range of difficulty. We then use the evolved data to train **EvolProver**, a 7B-parameter non-reasoning theorem prover. EvolProver establishes a new state-of-the-art (SOTA) on FormalMATH-Lite with a 53.8% pass@32 rate, surpassing all models of comparable size, including reasoning-based models. It also sets new SOTA records for non-reasoning models on MiniF2F-Test (69.8% pass@32), Ineq-Comp-Seed (52.2% pass@32), and Ineq-Comp-Transformed (34.0% pass@32). Ablation studies further confirm our data augmentation pipeline's effectiveness across multiple benchmarks.

## 1 Introduction

Large Language Models (LLMs) have demonstrated significant potential in mathematical reasoning, sparking a surge of research into their application for formal theorem proving. Formal languages like Lean (Moura & Ullrich, 2021), Coq (Barras et al., 1997), and Isabelle (Paulson, 1994) represent mathematical proofs as rigorous code implementations. This process demands strict syntactic precision and logical soundness, with every proof requiring compiler verification. While this guarantees the absolute reliability of proofs, it also creates a major bottleneck: the extreme scarcity of high-quality training data. Crafting formal proofs requires deep domain expertise and substantial time, a reality that fundamentally conflicts with the data-intensive paradigm of LLMs.

To address the scarcity for data, the research community has explored various data synthesis methods. For instance, DeepSeek-Prover (Xin et al., 2024a) attempts to automatically translate a large number of informal natural language problems into formal statements, using model scoring and a hypothesis

---

[*]   Equal contribution.

[†]   The idea of this work was proposed when the first author was an intern at Ant Group before joining HKBU, and the work was completed after he joined HKBU.

[‡]   Work done during internship at Ant Group.

[§]   Corresponding authors. Correspondence to: Jing Ma <majing@hkbu.edu.hk>, Da Zheng <zhengda.zheng@antgroup.com>, Lun Du <dulun.dl@antgroup.com>

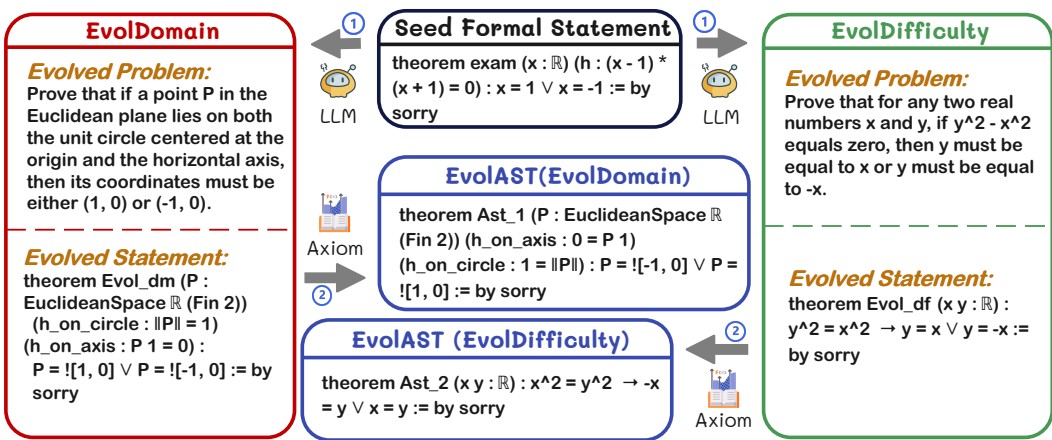

Figure 1: An example of problems evolved by EvolDomain, EvolDifficulty, and EvolAST. A seed formal statement is evolved in parallel by EvolDomain and EvolDifficulty, yielding two new statements. Each of these is then further evolved by EvolAST to generate syntactic variants.

rejection mechanism for screening. Goedel-Prover-V2 (Lin et al., 2025) adopts a scaffolded strategy to generate mathematical problems of appropriate difficulty to provide models with more effective learning signals. Meanwhile, STP (Dong & Ma, 2025) constructs two adversarial roles of a conjecturer and a prover that iteratively improve to jointly generate new problems and proofs.

However, a line of work has shown that models trained with such synthesized data still lack generalizability. For example, Zhao et al. (2025a) noted that minor transformations of a problem, such as transforming an inequality of the form $f(x) > g(x)$ to $f(x) + f(y) > g(x) + g(y)$, degrade the performance of LLMs drastically. Furthermore, other studies (Hao et al., 2025; Huang et al., 2025) have revealed that this fragility is not unique to formal reasoning; informal LLMs are also susceptible to minor problem transformations. Motivated by this, we propose a novel data augmentation pipeline to improve model generalizability by addressing it from two perspectives: symmetry and difficulty.

In mathematics, symmetry means exactly invariance under certain transformations. From the symmetry perspective, the fragility of existing models against minor transformations of problems suggests they fail to learn the underlying symmetry structure of the mathematical problem. To address this, we introduce two complementary methods targeting syntactic and semantic symmetry. The first, **EvolAST**, addresses syntactic symmetry using Abstract Syntax Tree (AST). It parses a formal statement into an AST, applies equivalence transformations using a library of axioms and theorems, and converts the modified tree back into a new statement. This generates semantically identical but syntactically diverse problems. The core strength of EvolAST is its extensibility, as any mathematical equivalence can be encoded as a new transformation rule, allowing for systematic enrichment of the data's structural diversity.

Our second method, **EvolDomain**, addresses semantic symmetry, where a theorem can be reinterpreted in different domains while preserving its core logic. EvolDomain uses evolution-based instructions to guide LLMs in translating theorems across mathematical domains, thereby creating novel and diverse problem statements.

From the difficulty perspective, studies have shown that models trained on data with a narrow difficulty range often fail to generalize (Jiang et al., 2023; Parashar et al., 2025). To mitigate this, we propose **EvolDifficulty**, a method that uses carefully designed instructions to evolve existing theorems by adjusting their difficulty. This process creates a dataset with a much broader difficulty spectrum, which discourages models from relying on shortcuts or mere memorization.

Combining EvolAST, EvolDomain, and EvolDifficulty, we create a comprehensive data augmentation pipeline. Example problems evolved by our pipeline in provided in Figure 1. We apply this pipeline to augment public datasets such as STP (Dong & Ma, 2025) and Deepseek-Prover-V1 (Xin et al., 2024a). By training DeepSeek-Prover-V1.5-Base on this augmented data, we produce our model, **EvolProver**. EvolProver achieves state-of-the-art (SOTA) performance on multiple benchmarks. Notably, EvolProver is a non-reasoning (i.e., non-CoT) model, yet it achieves results comparable to, and sometimes surpassing, those of reasoning models. On FormalMATH-Lite (Yu et al., 2025),

it sets a new SOTA with a 53.8% pass@32 rate among models of comparable size, including reasoning models. Furthermore, it establishes new SOTA pass@32 rates for non-reasoning models of comparable size on several benchmarks: 69.8% on MiniF2F-Test (Zheng et al., 2021), 52.2% on Ineq-Comp-Seed (Zhao et al., 2025a), and 34.0% on Ineq-Comp-Transformed (Zhao et al., 2025a). Ablation studies confirm the efficacy of our pipeline, showing that EvolProver outperforms its counterparts trained on unaugmented or partially augmented data, in some cases by over 10 percentage points.

The main contributions of this work can be summarized as follows:

- We propose a novel data augmentation pipeline that improves model generalizability by systematically enhancing formalized data directly from both symmetry and difficulty perspectives.

- We propose EvolAST, a highly extensible, AST-based method that generates syntactically diverse yet semantically equivalent problems by leveraging formal axioms and theorems as transformation rules. Additionally, we introduce EvolDomain and EvolDifficulty, two LLM-driven methods that enrich training data by translating problems across domains and evolving their difficulty, respectively.

- We train and release EvolProver, a powerful non-reasoning theorem prover built on our augmented data. EvolProver achieves state-of-the-art performance across multiple benchmarks, outperforming all comparable models on FormalMATH-Lite and setting new records for non-reasoning models on others.

## 2 RELATED WORKS

**Formal Provers.** Numerous LLM-based formal provers (Ji et al., 2025; Zhang et al., 2025; Shang et al., 2025) have emerged after the advent of ChatGPT, including reasoning-based models like DeepSeek-Prover-V2 (Ren et al., 2025), non-reasoning models like STP (Dong & Ma, 2025), and tree-search models like BFS-Prover-V1 (Xin et al., 2025a) & BFS-Prover-V2 (Xin et al., 2025b). Our work focuses on advancing the state-of-the-art for non-reasoning models, which offer significant computational efficiency.

**Data Augmentation in Mathematical Reasoning.** The critical need for large-scale, high-quality training data has spurred significant research into automated methods for mathematical problem generation. Prominent approaches in informal mathematics include MetaMath (Yu et al., 2024), which bootstraps new data by rewriting existing questions from multiple perspectives like rephrasing and backward reasoning. Similarly, WizardMath (Luo et al., 2025a) adapts the Evol-Instruct framework (Luo et al., 2025b; Xu et al., 2024) to systematically generate problems of varying complexity. Another work, PromptCoT (Zhao et al., 2025b), focuses on synthesizing complex problems by emulating the design process of human experts, grounding the generation in core mathematical concepts and logical structures. In the context of formal mathematics, STP (Dong & Ma, 2025) uses a single LLM that alternates between a "conjecturer" and a "prover" in a self-play loop, thereby synthesizing new formal conjectures and proofs from limited seed data. Ineq-Comp (Zhao et al., 2025a) starts from simple inequalities and applies small but systematic algebraic transformations, such as duplicating variables and adding or multiplying inequalities, or performing substitutions like squaring and taking square roots, to construct families of more structurally complex composite inequalities. Inspired by both these informal and formal data-augmentation pipelines, we introduce EvolDomain and EvolDifficulty. These methods also utilize LLMs but specifically focus on the evolution of formal mathematical statements to enhance their complexity and domain coverage, thereby increasing the diversity of the training data.

While these approaches expand the range and depth of generated problems, they also expose an inherent weakness of LLM-based evolution: the inevitable introduction of syntactic or semantic errors. To mitigate this issue, we propose EvolAST. EvolAST leverages the programmatic features of the Lean 4 proof assistant to perform rewrites directly at the Abstract Syntax Tree (AST) level. This approach ensures that all generated formal statements are syntactically correct and semantically equivalent, effectively increasing data diversity and precision.

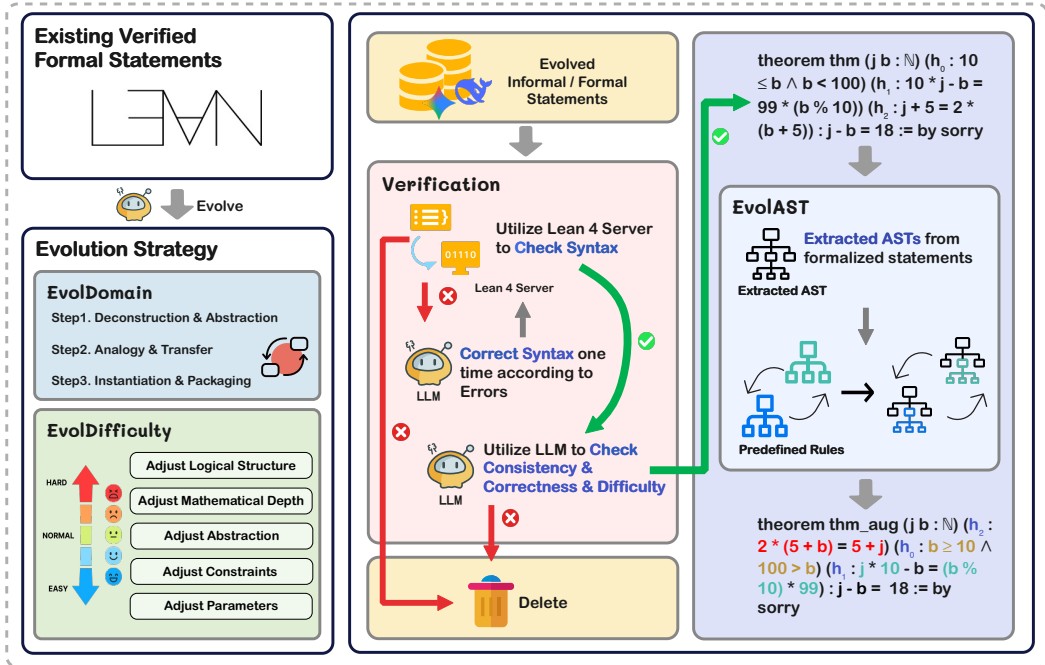

Figure 2: The workflow of our data augmentation pipeline comprises three phases: EvolDomain and EvolDifficulty, Verification, and EvolAST.

**Robustness of LLMs in Mathematical Reasoning.** Recent work has highlighted that LLMs lack robustness against small perturbations in mathematical problems, such as variable renaming or adding noise. For instance, the PutnamGAP benchmark (Hao et al., 2025) tests equivalence-preserving variants and shows average accuracy declines of 3-10%. Similarly, MATH-P-Hard (Huang et al., 2025) introduces structural shifts, causing performance drops of 10-25% in models like o1-mini.

While this issue is recognized in informal mathematics, the robustness of LLMs in formal reasoning systems like Lean 4 and Coq remains largely underexplored. Built from systematically composed inequalities, the Ineq-Comp benchmark (Zhao et al., 2025a) was developed to address this gap by measuring a prover's performance drop between original problems and their perturbed counterparts.

## 3 METHOD

Our methodology is centered around a multi-stage data augmentation pipeline, as illustrated in Figure 2. First, we leverage LLMs to expand existing formal statements through two evolution-based processes: **EvolDomain**, for cross-domain translation, and **EvolDifficulty**, for complexity adjustment. After a rigorous verification stage, we further diversify the data's syntactic structure using **EvolAST**, a deterministic AST-based transformation method. Finally, we train our model, **EvolProver**, on this augmented dataset. To control the instability introduced at each stage, we note that the LLM-driven steps (EvolDomain, EvolDifficulty) behave like exponential magnification ($x \to \exp(x)$): since they are based on LLMs, they may introduce significant and unpredictable changes, and the propagation of such instability can be likened to an exponential function where a small initial perturbation $x$ may grow into $\exp(x)$. In contrast, EvolAST, as a deterministic, syntax-based transformation, introduces much smaller instability; its effect is more similar to a linear function, effectively turning $x$ into $2x$. Executing the LLM-based stages first and then applying the syntactic transformation ($x \to \exp(x) \to 2\exp(x)$) avoids prematurely amplifying noise. Keeping the overall instability at a moderate level is crucial, because overly unstable problems become hard to prove, which in turn prevents us from obtaining enough valid training instances to support an effective data augmentation strategy. This procedure allows us to both enrich the diversity of the data and maintain a sufficient amount of training data. The following sections detail each component of this pipeline.

### 3.1 EVOLDOMAIN AND EVOLDIFFICULTY

Proven formal statements, with their inherent semantic and syntactic correctness, serve as ideal seeds for data generation. Our work mainly builds upon two open-source datasets, Deepseek-Prover-V1 and STP-Lean, which often lack natural language descriptions. We therefore evolve these formal statements directly by instructing an LLM to generate new, related theorems. This approach leverages the logical structure embedded in the formal language itself as a basis for creative generation, bypassing the need for natural language intermediaries.

**EvolDomain.** EvolDomain leverages an LLM to translate a formal statement into new mathematical domains. This process involves three main steps: 1) abstracting the statement's logical skeleton, 2) identifying an analogous concept in a target domain, and 3) instantiating a new, concrete proposition based on this analogy.

Formally, let this process be a function $\mathcal{F}$. Given a source statement $S_i^{\text{formal}}$ and a target domain $D_m$ (selected from a predefined list $\mathcal{L}_D = \{D_1, D_2, \ldots, D_M\}$), $\mathcal{F}$ guides an LLM to first extract the statement's abstract logical skeleton. Based on this skeleton, the model identifies a structurally similar concept in $D_m$ and uses it to construct a new proposition. The output is a pair consisting of a natural language description, $\widehat{P}_i$, and a new formal statement, $\widehat{S}_i^{\text{formal}}$. This can be formally represented as: $\mathcal{F}(S_i^{\text{formal}}, D_m) = (\widehat{S}_i^{\text{formal}}, \widehat{P}_i)$.

To maximize the exploration of logical connections across domains, our prompt further guides the LLM to simultaneously transfer and instantiate the core logical skeleton into 3 to 5 distinct new domains. Therefore, the final output of a single function call is a set of pairs spanning multiple domains, with each pair containing a new formal statement and its corresponding natural language description. Prompt templates and examples can be found in Appendix A.3.

**EvolDifficulty.** EvolDifficulty leverages an LLM to adjust a formal statement's difficulty, thereby creating a dataset with a broad difficulty spectrum. We denote this process by the function $\mathcal{E}$. The process, $\mathcal{E}$, is guided by carefully designed evolution strategies. Based on expert consultation, we designed five core evolution strategies, $\mathcal{S} = \{s_1, \ldots, s_5\}$: (1) Adjusting Logical Structure, (2) Adjusting Mathematical Depth, (3) Adjusting Abstraction, (4) Adjusting Constraints, and (5) Adjusting Parameters. Given a formal statement $S_i^{\text{formal}}$, the function applies a strategy $s_k \in \mathcal{S}$ with an evolution direction $\delta \in \{+1, -1\}$ (for increasing or decreasing difficulty, respectively) to instruct an LLM to generate a new pair of a new formal statement $\widehat{S}_i^{\text{formal}}$ and its natural language description $\widehat{P}_i$. This can be formally represented as $\mathcal{E}(S_i^{\text{formal}}, s_k, \delta) = (\widehat{S}_i^{\text{formal}}, \widehat{P}_i)$.

By systematically applying this framework, EvolDifficulty enables fine-grained control over dataset difficulty, generating problems with a smooth gradient that enriches the dataset's hierarchical structure. Prompt templates and examples can be found in Appendix A.4.

**Verification.** We employ a stringent two-stage verification pipeline to ensure data quality. First, each generated statement $\widehat{S}i^{\text{formal}}$ is validated for syntactic integrity using the Lean 4 compiler. Statements that fail are given a single LLM-based repair attempt before being discarded. Second, all syntactically valid pairs $(\widehat{S}_i^{\text{formal}}, \widehat{P}_i)$ undergo semantic evaluation by an LLM-based judge. The judge assesses three aspects: consistency between the formal and natural language versions, propositional correctness, and difficulty appropriateness. This dual-filter mechanism, combining deterministic compilation with semantic judgment, ensures that only syntactically sound and semantically coherent data populates our final dataset. Prompt templates can be found in Appendix A.5.

### 3.2 EVOLAST

EvolAST is founded on the principle that formal language statements, as structured code, can be parsed into Abstract Syntax Tree (AST). This allows us to bypass non-deterministic models and instead apply a deterministic set of rewriting rules based on established axioms and theorems, guaranteeing semantic equivalence.

We formalize this process as a function $\mathcal{A}$. EvolAST implements an extensible set of rewriting rules (currently 7 rules), $\mathcal{R} = \{r_1, \ldots, r_7\}$, where each rule $r_k$ corresponds to a specific logical equivalence: (1) Hypothesis Reordering, (2) Commutativity, (3) Associativity, (4) Distributivity, (5)

De Morgan's Laws, (6) Operand Swapping for Symmetric Relations, and (7) Dual Relation Conversion. Given an input statement $S_i^{\text{formal}}$, the function $\mathcal{A}$ first parses it into an AST. It then recursively traverses the tree, applying any applicable rule $r_k \in \mathcal{R}$ at each node with a predefined probability $p$. Finally, the modified AST is recompiled into a new formal statement $\widehat{S}_i^{\text{formal}}$. The process can be formally represented as $\mathcal{A}(S_i^{\text{formal}}, p) = \widehat{S}_i^{\text{formal}}$. We provide an example in Appendix A.6.

Since all transformations are based on strict logical equivalences, EvolAST generates syntactically diverse data while ensuring semantic correctness, thus eliminating the need for further verification. The framework is highly extensible, as any known mathematical or logical equivalence can be encoded as a new rewriting rule.

### 3.3 TRAINING EVOLPROVER

We train our final model, EvolProver, by fine-tuning DeepSeekProver-V1.5-Base (Xin et al., 2024b) on our augmented dataset. DeepSeekProver-V1.5-Base is currently the strongest model pretrained purely on large-scale formal theorem-proving data without chain-of-thought supervision, making it particularly well suited for our synthetic data and our goal of enhancing fast, non-reasoning provers. The training process consists of two stages: Supervised Fine-Tuning (SFT) and Reinforcement Learning (RL). Detailed information on dataset curation and training algorithms can be found in Appendix A.1.

For comparison and ablation studies, we also trained several other models. This includes a baseline model, **EvolProver-Base**, which was trained exclusively on the original, unaugmented public data. We also prepared a series of specialized models for our comprehensive ablation experiments, with details provided in Appendix A.2.

## 4 EXPERIMENTS

### 4.1 BASELINES

Existing formal provers are broadly categorized into three types: non-reasoning, reasoning, and tree-search models.

**Non-reasoning models** generate proofs end-to-end without an intermediate thought process. Key examples include DeepSeek-Prover-V2 (non-CoT) (Ren et al., 2025), Goedel-Prover-SFT (Lin et al., 2025), and STP (Dong & Ma, 2025).

**Reasoning models** employ a chain-of-thought process to generate proofs, where the reasoning process is often significantly longer than the final proof. Key examples are DeepSeek-Prover-V2 (CoT) (Ren et al., 2025), Moonshot's Kimi-Prover-Preview (Wang et al., 2025a) and Kimi-Prover (Wang et al., 2025b), and Goedel-Prover-V2 (Lin et al., 2025). Notably, DeepSeek-Prover-V2 has both a reasoning and a non-reasoning mode. While generally higher performing, reasoning models demand substantial computational resources due to their chain-of-thought approach (e.g., more than 6000 tokens per proof vs. less than 700 for non-reasoning models). This focus on token efficiency has spurred a recent wave of interest in fast, non-reasoning models, such as Claude 4's Non-thinking mode (Anthropic, 2025) and Grok-Code-Fast-1 (xAI, 2025).

**Tree-search models** represent an intermediate proof state as a node in a search tree and use a model to assign heuristic scores to guide the search order. Key examples include BFS-Prover (Xin et al., 2025a), DeepSeek-Prover-V1.5 + RMaxTS (Xin et al., 2024b), and InternLM2.5-StepProver (Wu et al., 2024).

For our comparative analysis, we report the performance metrics as published by the original authors to ensure consistency and avoid discrepancies from our own re-evaluations.

### 4.2 RESULTS

**FormalMATH** (Yu et al., 2025) is a broad dataset of formal theorems. We follow standard practice and evaluate on its 425-problem subset, FormalMATH-Lite, as other problems in the full dataset were

Table 1: Comparison with SOTA 7B-size models on the FormalMATH-Lite dataset; ↑ means increase in absolute performance over the ablation model EvolProver-Base; Average Token Length means the average number of output tokens across the benchmark. We do not report average token length for tree-search models, as this metric is not directly comparable with other model types.

| Models | Average Token Length | Sample Budget | FormalMATH |
|---|---|---|---|
| *Reasoning Models* | | | |
| DeepSeek-Prover-V2(COT) | 4804.6 | 32 | 51.76% |
| Kimina-Prover-Preview | 6097.7 | 32 | 48.94% |
| *Tree-Search Models* | | | |
| InternLM2.5-StepProver | N/A | $1 \times 3200$ | 7.87% |
| BFS-Prover | N/A | $1 \times 3200$ | 27.19% |
| *Non-Reasoning Models* | | | |
| DeepSeek-Prover-V1.5-SFT | 115.9 | 32 | 40.40% |
| DeepSeek-Prover-V1.5-RL | 163.4 | 32 | 47.98% |
| Goedel-Prover-SFT | 458.4 | 32 | 46.70% |
| STP | 186.8 | 32 | 48.59% |
| **DeepSeek-Prover-V2-7B-non-CoT** | 394.5 | 32 | **50.35%** |
| **EvolProver-Base(Ours)** | 629.8 | 32 | **44.71%** |
| **EvolProver(Ours)** | 653.7 | 32 | **53.86%**(↑ 9.15%) |

used in training. Problems within FormalMATH-Lite were held out and used exclusively for final evaluation.

The results are summarized in Table 1. EvolProver achieves a new SOTA of 53.86% among models of comparable size, surpassing the previous best of 51.76%. Notably, our non-reasoning model outperforms top reasoning models like DeepSeek-Prover-V2 and Kimi-Prover-Preview. Furthermore, EvolProver outperforms its baseline, EvolProver-Base, by 9.15 percentage points, demonstrating the significant impact of our data augmentation pipeline.

**MiniF2F** (Zheng et al., 2021) is a standard benchmark comprising 488 problems from mathematics competitions. Following common practice, we report results on its 244-problem test set, MiniF2F-Test. The results are presented in Figure 3. EvolProver achieves a pass@32 rate of 69.80% on MiniF2F-Test, establishing a new SOTA performance among non-reasoning models of comparable size. Notably, this performance is comparable to, and in some cases exceeds, that of reasoning models, despite using significantly fewer tokens (a nearly 10-fold reduction in token consumption). Due to the construction of the training data, the content generated by EvolProver, in addition to the proof, also includes a restatement of the problem, while further analysis is placed in section A.8, which makes its outputs slightly longer than those of the other non-chain-of-thought models.

**Ineq-Comp** (Zhao et al., 2025a) is a benchmark designed to evaluate the robustness of formal provers against minor problem perturbations. It contains 75 seed problems from Olympiad-level inequalities and 150 corresponding transformed variants. Each seed problem is systematically altered through simple operations(e.g., algebraic rewrites, variable duplication) to create two transformed variants. While humans can easily solve these transformed problems, formal provers often struggle with them even if they can solve the original. A model's robustness is measured by the ratio of its performance on transformed problems to its performance on the seed problems, for which a higher ratio indicates greater robustness.

Our results are presented in Table 2. EvolProver again sets a new SOTA for non-reasoning models on all three metrics (seed, transformed, and ratio), outperforming the next-best non-reasoning model by a significant margin. Its performance is also comparable to that of top reasoning models. Notably, our data augmentation pipeline leads to a substantial boost in robustness: EvolProver's robustness ratio is 30.74 percentage points higher than that of EvolProver-Base, demonstrating the effectiveness of our approach.

## 5 ANALYSIS

**Evolution Strategy.** EvolDifficulty and EvolDomain employ a general LLM to directly evolve formalized mathematical theorems. This approach addresses the inherent complexity of mathematical

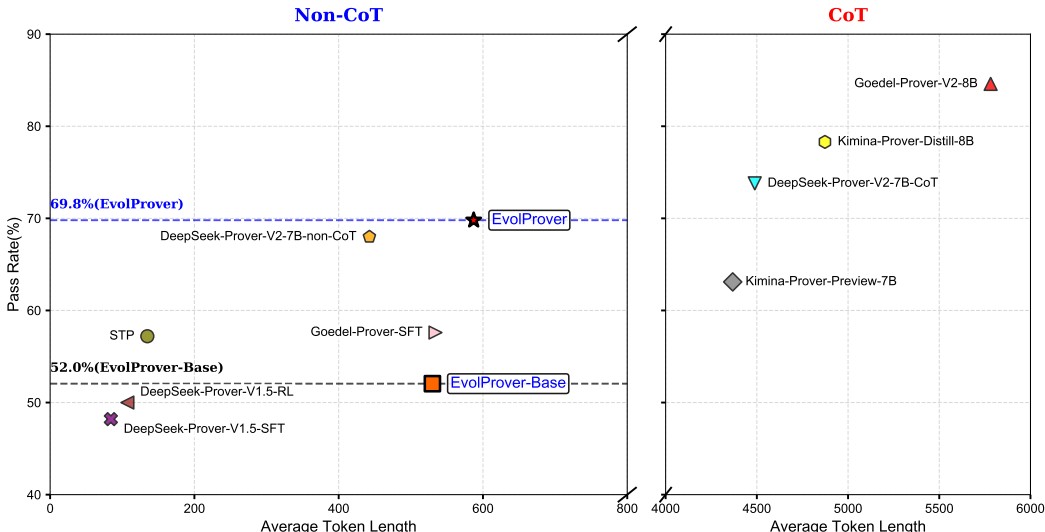

Figure 3: Comparison with SOTA models on the MiniF2F-Test dataset. Pass Rate means pass@32 success rate. Average token length is the average number of tokens generated by models across the benchmark. We categorize models as Non-CoT(non reasoning) and CoT(reasoning)

Table 2: Comparison with SOTA 7B-size models on the Ineq-Comp Benchmark; Pass means pass@32 rate for reasoning models and non-reasoning models, and means $1 \times 3200$ pass rate for tree-search models.↑ means increase in absolute performance over the ablation model EvolProver-Base.

| Models | Pass on Seed | Pass on Transformed | Pass Ratio |
|---|---|---|---|
| *Reasoning Models* | | | |
| DeepSeek-Prover-V2 (COT) | 66.23% | 44.53% | 67.23% |
| Kimina-Prover-Preview | 50.06% | 27.58% | 55.09% |
| *Tree-Search Models* | | | |
| DeepSeek-Prover-V1.5 (RL + RMaxTS) | 42.66% | 14.83 % | 34.76 % |
| InternLM2.5-StepProver | 25.59% | 3.44 % | 16.6 % |
| *Non-Reasoning Models* | | | |
| DeepSeek-Prover-V1.5 (RL) | 34.40% | 6.68% | 19.42% |
| Goedel-Prover-SFT | 43.46% | 14.54% | 33.47% |
| DeepSeek-Prover-V2-7B-non-CoT | 56.00% | 27.33% | 48.90% |
| **EvolProver-Base (Ours)** | **43.26%** | **14.89%** | **34.43%** |
| **EvolProver (Ours)** | **52.20%**(↑ 8.94%) | **34.02%**(↑ 19.13%) | **65.17%**(↑ 30.74%) |

formalization(Lu et al., 2025; Li et al., 2025), a task traditionally reliant on specialized models trained to convert natural language problems into formal expressions. However, the direct application of general-purpose LLMs for this purpose remains relatively unexplored, leaving their comparative advantages and limitations as an open question.

To validate our strategy of directly evolving formal statements, we compare it against a common alternative: evolving Natural Language (NL) problems first and then formalizing them. We designed a controlled experiment with four branches:

- **EvolDomain & EvolDifficulty (Ours)**: Directly evolves new formal statements from existing ones.
- **Formalization-Formalizer**: Evolves NL problems, then formalizes them using a specialized model (Kimina-Formalizer-7B).
- **Formalization-LLM-zero-shot**: Evolves NL problems, then formalizes them using a general-purpose LLM (Gemini-2.5-Pro) in a zero-shot setting.
- **Formalization-LLM-few-shot**: The same as above, but in a few-shot setting.

Starting with 400 seed problems, we generated an equal number of candidates using each method and passed them through our stringent verification pipeline. The number of successfully verified

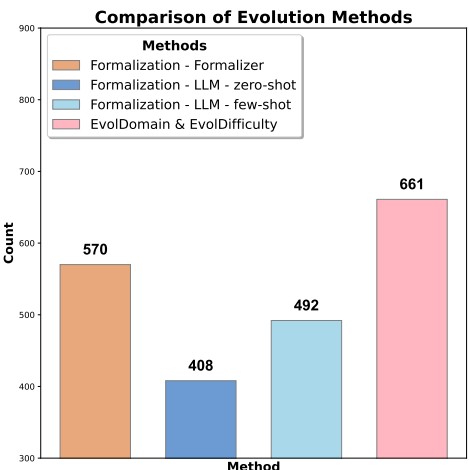

Figure 4: Comparison of the number of candidates passing verification for four evolution methods. Our EvolDomain & EvolDifficulty performs best.

statements for each method is shown in Figure 4. Our direct evolution approach significantly outperforms all NL-based methods, confirming its superiority. The final candidate count can exceed 400 as each seed may yield multiple valid variants. Additionally, we directly evaluated the semantic quality of 1,634 data with EvolDomain and EvolDifficulty using DeepSeek-V3.1. Among them, 496 statements failed verification, corresponding to a semantic failure rate of 30.35%.

**Domain Diversity.** Here, we analyze how our framework improves domain diversity and how this enhancement translates to performance gains. Figure 5 illustrates the effect of EvolDomain on a sample of 200 seed problems. The initial distribution is heavily skewed, with domains like Algebra dominating while others like Calculus are absent. After applying EvolDomain, the dataset becomes significantly more balanced: the share of over-represented domains is reduced, and previously missing categories are introduced. The domains for both sets were classified by DeepSeek-V3 and human-verified.

This improved diversity directly leads to better model performance across various domains, as detailed in Table 3. Comparing EvolProver against the EvolProver-Base baseline, our full model achieves gains across most categories. Critically, it makes a breakthrough in Calculus, solving 3 problems where the baseline solved 0. These results confirm that our strategy not only enriches domain diversity but also enhances the model's overall mathematical capabilities.

**Similarity.** Additionally, we investigate whether the evolution process leads to the training data containing more samples similar to the test sets. To this end, we conduct an analysis on the MiniF2F-Test, FormalMATH-Lite, and Ineq-Comp test sets. We use each test sample as a query and employ the state-of-the-art embedding model Qwen3-Embedding-8B to retrieve its top-1 most similar sample

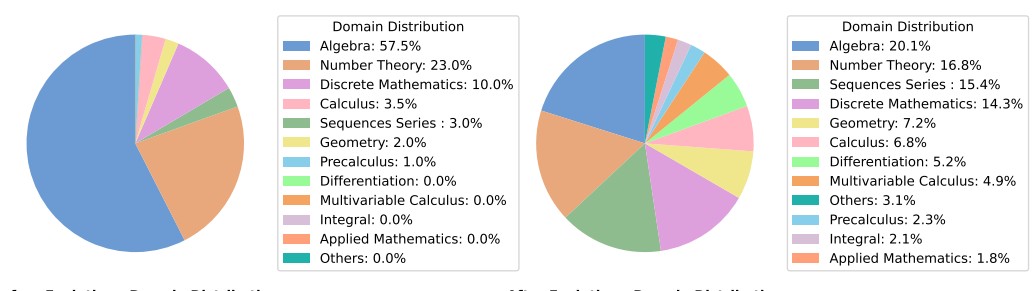

Figure 5: Comparison of Mathematical Domain Distribution Before and After EvolDomain.

Table 3: Number of proved problems on FormalMATH-Lite benchmark in different domains under 32 generation trials. EvolProver improves upon EvolProver-Base across most domains.

| Domain | EvolProver-Base | EvolProver | Total |
|---|---|---|---|
| Algebra | 121 | 141 (+20) | 235 |
| Applied Mathematics | 28 | 33 (+5) | 46 |
| Number Theory | 16 | 23 (+7) | 45 |
| Precalculus | 14 | 15 (+1) | 23 |
| Geometry | 7 | 8 (+1) | 17 |
| Discrete Mathematics | 2 | 5 (+3) | 25 |
| Calculus | 0 | 3 (+3) | 6 |
| Multivariable Calculus | 2 | 2 (−) | 6 |
| Others | 0 | 0 (−) | 23 |

Table 4: Ablation experiment results on the FormalMATH-Lite benchmark, the MiniF2F-Test benchmark, and the Ineq-Comp benchmark. All results are pass@32 rate. Superscripts denote the training data used: superscript$^0$ for Public dataset only; superscript$^{0+1}$ for Public dataset + EvolDomain & EvolDifficulty augmentation; superscript$^{0+1+2}$ for Full augmentation including Public dataset, EvolDomain & EvolDifficulty, and EvolAST. EvolProver-Ablation-SFT and Evoler-SFT are trained through a sole SFT stage. EvolProver-Base, EvolProver-Ablation-RL and EvolProver are trained through an SFT stage and an RL stage.

| Models | FormalMATH | MiniF2F | Ineq-Comp (Seed) | Ineq-Comp (Transformed) | Ineq-Comp (Ratio) |
|---|---|---|---|---|---|
| EvolProver-Base$^0$ | 44.71% | 52.05% | 43.26% | 14.89% | 34.43% |
| EvolProver-Ablation-SFT $^{0+1}$ | 50.35% | 65.16% | 49.79% | 29.19% | 58.62% |
| EvolProver-SFT $^{0+1+2}$ | 51.53% | 66.39% | 49.82% | 30.35% | 60.19% |
| EvolProver-Ablation-RL $^{0+1}$ | 51.98% | 68.22% | 50.36% | 33.05% | 65.62% |
| **EvolProver** $^{0+1+2}$ | **53.96%** | **69.80%** | **52.20%** | **34.02%** | **65.17%** |

from our final training data. These paired theorems are then evaluated by DeepSeek-V3.1 for human-like similarity assessment. We instruct the model to score each pair on a scale from 1 to 10, where 1 denotes "completely dissimilar" and 10 denotes "semantically identical." The evaluation results show that the average similarity score of these pairs is about **3.48**, indicating that even among the most syntactically and semantically similar samples between the training and test sets, the overall similarity remains quite limited. Further details are provided in Appendix A.7.

**Ablation Experiments.** To further validate the effectiveness of our proposed methods, we conduct a series of comprehensive ablation studies. The results are presented in Table 4. These experiments isolate the impact of each component and demonstrate that they provide consistent benefits across multiple benchmarks and at various training stages. In addition, we analyze the mathematical domains of the problems in the FormalMATH-Lite benchmark based on these ablation experiments. Experimental details are provided in Appendix A.2.

## 6 CONCLUSION AND FUTURE WORK

In this paper, we introduced a highly-extensible data augmentation pipeline with three methods: EvolDomain, EvolDifficulty, and EvolAST, designed to improve model generalizability from semantic and syntactic perspectives. Our resulting model, EvolProver, achieves new SOTA results on several key benchmarks, notably surpassing all comparable models on FormalMATH-Lite. For future work, we plan to enhance EvolProver's reasoning capabilities by incorporating synthetically generated Chain-of-Thought data into its training.

## REPRODUCIBILITY STATEMENT

We are strongly committed to the reproducibility of our work. Our EvolAST method is designed to be highly extensible, and we encourage the community to contribute by expanding its set of applicable axioms and theorems.

## ACKNOWLEDGMENT

This work was supported by Ant Group Research Intern Program, and National Natural Science Foundation of China Young Scientists Fund (No. 62206233).

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

# A APPENDIX

## A.1 DETAILS OF TRAINING EVOLPROVER

**Data Curation.** Our data curation process follows a multi-stage funnel. We begin with a seed pool of approximately 3.3 million verified formal statements aggregated from four sources: DeepSeek-Prover-V1, STP-lean, MiniF2F-Valid, and FormalMATH-All (excluding the FormalMATH-Lite subset).

From this pool, we sample 70k statements for evolution. These initial 70k samples are carefully curated from our four sources. Three of the datasets—DeepSeek-Prover-V1, FormalMATH-All (excluding FormalMATH-Lite), and MiniF2F-Valid—are high-quality but relatively small, collectively containing just over 30,000 theorems, and we include all theorems from these valuable sources in our synthesis pool. The fourth source, STP-lean, is a much larger synthetic dataset with 3.26 million theorems; to prevent this synthetic data from dominating the more curated sources and to avoid excessive computational costs, we first filter out theorems originating from `mathlib` and then perform a weighted sampling over the remainder to select an additional 30,000+ examples, guided by the importance weights provided by the STP-lean authors. The number of sampled examples is chosen to be comparable to the total size of the other real-world datasets combined, yielding a high-quality and diverse seed set of 70k theorems in a cost-effective manner, balancing valuable real-world problems with a broader set of synthetic examples. These are processed by our EvolDomain and EvolDifficulty methods using Gemini-2.5-Pro and DeepSeek-R1, which then undergo a verification process to yield 57.4k high-quality (statement, description) pairs. This verification first involves a syntax check using the Lean 4 compiler; if the compiler finds a syntax error, we use DeepSeek-V3 (DeepSeek-AI, 2025) to repair it, after which DeepSeek-V3 performs a final semantic check. Next, we apply EvolAST to this set for syntactic diversification, expanding it to approximately 96.7k entries (as a single statement can generate multiple AST variants). Finally, we generate proofs for each statement. Using DeepSeek-Prover-V2-671B and Goedel-Prover-V2-8B as expert models, we generate 50 proof candidates per statement and retain only those that pass Lean 4 compiler verification. After removing duplicates, this process results in a final training dataset of 39.2k unique (statement, proof) pairs. Thus, our 39.2k training data are synthesized from this carefully curated 70k-theorem seed set. To prevent data leakage, we ensure that the initial states of all theorem statements in our data are different from those in the tested benchmarks.

**Token Cost.** Our data construction can be divided into four stages: EvolDomain & EvolDifficulty, Verification process, EvolAST, and Proof synthesis. We counted the token cost for each stage as follows:

| Stage | Token Consumption |
|---|---|
| Problem synthesis (EvolDomain & EvolDifficulty) | 893M |
| Verification process | 506M |
| EvolAST | 0 |
| Proof synthesis | 11B |

**Supervised Fine Tuning.** We fine-tune the DeepSeek-Prover-V1.5-Base model using full-parameter supervised fine-tuning (SFT). Our training data includes approximately 3.3M theorems with proofs from public datasets and 39.2K theorems with proofs from the augmented datasets. The model is trained for one epoch with the AdamW optimizer. We set the initial learning rate to $1.0 \times 10^{-5}$ and decay it using a cosine scheduler with a 5% warmup ratio. All sequences are truncated to a maximum length of 4096 tokens, and we use a global batch size of 32.

**Reinforcement Learning.** Following the Supervised Fine-Tuning (SFT) stage, we further enhance the model's performance by applying Reinforcement Learning (RL) to the SFT checkpoint. For RL training, For RL training, we use augmented data (39.2K) and additionally incorporate FormalMATH-All (excluding FormalMATH-Lite) and MiniF2F-Valid into our training data. We employ a standard binary reward: for each problem, the model receives a reward of 1 if the generated Lean proof is correct, and 0 otherwise. This RL fine-tuning process produces the final EvolProver.

Table 5: Domain-wise performance of EvolProver variants.

| Domain | EvolProver-Base | EvolProver-Ablation-SFT | EvolProver-SFT | EvolProver-Ablation-RL | EvolProver |
|---|---|---|---|---|---|
| Algebra | 121 | 133 | 134 | 133 | 141 |
| Applied Mathematics | 28 | 30 | 32 | 32 | 33 |
| Number Theory | 16 | 21 | 23 | 23 | 23 |
| Precalculus | 14 | 15 | 15 | 15 | 15 |
| Geometry | 7 | 8 | 8 | 8 | 8 |
| Discrete Mathematics | 2 | 3 | 3 | 5 | 5 |
| Calculus | 0 | 2 | 2 | 3 | 3 |
| Multivariable Calculus | 2 | 2 | 2 | 2 | 2 |
| Others | 0 | 0 | 0 | 0 | 0 |

**RL Training Details**    To improve training efficacy, we curate the RL training dataset by filtering problems based on the pass@1 success rate of the SFT checkpoint. We include only problems where $0 < \text{pass@1} < 1/2$. This selection strategy ensures that the training set is challenging yet solvable for our model. The filtered dataset contains 2,718 problems. We initialize both the actor and critic models with the weights from the SFT checkpoint and train them using Proximal Policy Optimization (PPO). The training runs for 10 epochs with a batch size of 256, a constant actor learning rate of $1.0 \times 10^{-6}$, a constant critic learning rate of $1.0 \times 10^{-5}$, a clip ratio of 0.2, and a KL divergence loss coefficient of 0.001.

## A.2    ABLATION EXPERIMENTS

### A.2.1    ABLATION MODEL TRAINING

To precisely evaluate the contribution of each component, we trained a series of ablation models under controlled conditions. All training hyperparameters were kept identical across corresponding stages. The models are:

- **EvolProver-Base**: Our baseline, trained on the original, unaugmented public dataset through both SFT and RL stages.

- **EvolProver-Ablation-SFT**: Trained on data augmented only by EvolDomain and EvolDifficulty, and only undergoes the SFT stage.

- **EvolProver-Ablation-RL**: Same data as above (EvolDomain and EvolDifficulty only), but undergoes the full SFT and RL training process. This model directly isolates the impact of EvolAST when compared to the final EvolProver.

- **EvolProver-SFT**: The checkpoint of our final model after being trained on the fully augmented dataset (including EvolAST) for the SFT stage only.

We did not create an "EvolAST-only" model, as EvolAST operates on the output of EvolDomain and EvolDifficulty, making such an experiment logically infeasible.

### A.2.2    ABLATION EXPERIMENT RESULTS

The results of our ablation experiments, presented in Table 4, lead to two key conclusions. First, data augmentation provides a substantial boost, with even the partially augmented models (Ablation-SFT/RL) drastically outperforming the EvolProver-Base model across all benchmarks, often by more than 10%. Second, the EvolAST method consistently yields further improvements across all benchmarks. In the SFT stage, EvolProver-SFT (with EvolAST) surpasses EvolProver-Ablation-SFT (without EvolAST). Similarly, in the RL stage, the final EvolProver outperforms EvolProver-Ablation-RL. This demonstrates the value of the EvolAST method in both training phases.

To better understand how each augmentation component affects performance across mathematical domains, we conduct a fine-grained analysis and report the results in Table 5. This breakdown allows us to examine the domain-specific behaviors of different model variants and how they interact with data scale and difficulty. The detailed analysis is as follows:

- For EvolProver-Ablation-SFT, which retains only EvolDomain and EvolDifficulty, performance improves over EvolProver-Base in almost all domains. The relative gains differ

across domains and correlate with data scale. In long-tail domains with relatively sparse training data, the main improvements are associated with EvolDomain: by performing cross-domain semantic evolution, it transfers structural patterns from high-resource domains and effectively increases coverage in low-resource domains. In high-resource domains where the base model already captures the main solution patterns, EvolDifficulty contributes more prominently: by broadening the difficulty distribution of training problems, it encourages the model to handle a wider range of problem difficulty. Overall, these results indicate that EvolDomain primarily mitigates data scarcity in low-resource domains, while EvolDifficulty mainly enhances robustness to varying difficulty levels in high-resource domains, and the two components play complementary roles across different data scales.

- EvolProver-SFT is obtained by further incorporating EvolAST on top of EvolProver-Ablation-SFT. Relative to Ablation-SFT, which uses only EvolDomain and EvolDifficulty, EvolProver-SFT yields small but consistent additional gains in domains with a larger proportion of symbolic reasoning, such as algebra, number theory, and applied mathematics (for example, Algebra improves from 133 to 134, Applied Mathematics from 30 to 32, and Number Theory from 21 to 23), while performance in other domains remains essentially unchanged. This pattern suggests that EvolAST does not primarily introduce new semantic content. Instead, by generating multiple equivalent structured representations for the same reasoning process, EvolAST strengthens the model's ability to handle different surface forms with the same underlying meaning, leading to additional generalization benefits in high-resource domains that already have substantial sample size and semantic diversity.

- EvolProver-Ablation-RL is produced by applying reinforcement learning on top of EvolProver-Ablation-SFT, using data generated by EvolDomain and EvolDifficulty. Compared with EvolProver-SFT, performance in high-resource domains (such as Algebra, Applied Mathematics, and Number Theory) remains largely stable, whereas additional gains appear in sparser, more difficult, or larger-search-space domains (for example, Discrete Mathematics increases from 3 to 5 and Calculus from 2 to 3). These observations indicate that, under a data distribution shaped by cross-domain coverage and difficulty evolution, reinforcement learning can make fuller use of the evolved samples through exploratory optimization, particularly in domains where search and exploration are more challenging.

- The final EvolProver model performs reinforcement learning on the full dataset that includes all three augmentations: EvolAST, EvolDomain, and EvolDifficulty. Relative to EvolProver-Ablation-RL, EvolProver achieves further improvements in high-resource domains such as Algebra (from 133 to 141) and Applied Mathematics (from 32 to 33), while preserving the gains already observed in Discrete Mathematics (5) and Calculus (3). Taken together, these results support the view that the three augmentation components act along complementary axes—domain coverage, difficulty distribution, and structural diversity—and that combining them within both supervised fine-tuning and reinforcement learning leads to more robust and broadly improved domain-wise performance.

## A.3 DETAILS FOR EVOLDOMAIN

### A.3.1 PROMPT TEMPLATE

Formal problems can precisely extract the universal logical skeleton of a mathematical problem(Cao et al., 2025; 2026; Zheng et al., 2025). Our strategy leverages this by transferring that structure to new domains to systematically create rigorous new problems. Our preset domains cover a range of topics from high school competition problems to undergraduate-level subjects. The prompt template format of EvolDomain is as follow:

---

**Prompt Template for EvolDimain**

Your task is to start with a given Lean 4 formalized problem and follow the strategy below to formulate a new problem in a different mathematical domain.

### Transformation Strategy

Step 1. Deconstruction & Abstraction
Identify the original statement's abstract logical skeleton by isolating its core components. This involves recognizing the underlying mathematical objects, the essential operations being performed, and the fundamental relationship being asserted.

Step 2. Analogy & Transfer

Find a parallel structure in a new mathematical domain by identifying an analogous sequence of objects in the list below.
["Algebra", "Number Theory", "Integral", "Precalculus", "Differentiation", "Multivariable Calculus", "Sequences Series", "Applied Mathematics", "Discrete Mathematics", "Geometry", "Calculus", "Other"]
Then, translate the original operations and relations into concepts that are natural within this new context.

Step 3. Instantiation & Packaging

Formulate a new, concrete problem by applying the transferred structure and operations to the analogous objects. Package this new proposition into a clear natural language description and a rigorous formal statement.

Evolved variants should follow the following requirements:
1. Each variant should be mathematically rigorous and meaningful
2. Each variant should be syntax correct and a valid Lean 4 statement
3. Each variant should differ from the original mathematical domain
4. Each variant should follow the same output format as "### Example Variant Format"

### Example Variant Format:
```NL Description
Prove that ...
```
```Formal Statement
... := by sorry
```
``` Domain
Target Domain in the list
```

Please provide 3-5 variants following the strategies and requirements above, based on original statement.

---

```
### Original Formal Statement
```lean4
{Original Formal Statement}
```

### Evolution Results (3-5 variants in other mathematical domains)
```

### A.3.2 CASE STUDY

We select an example of EvolDomain from the evolved dataset. The original Lean 4 statement is as follows:

> **Original Lean 4 Statement (Number Theory)**
>
> ```
> theorem lean_workbook_12011 (m, n : ℤ) (h_1 : 2*m + n = 0) : m^3 ≠ n^3 - 15 := by
>   sorry
> ```

The evolved Lean 4 statement and its corresponding natural language description are as follows:

> **Evolved Statement and its corresponding Natural Language Question (Geometry)**
>
> Evolved Formal Statement:
> ```
> theorem affine_points_analogy P Q R : EuclideanSpace ℝ (Fin 2)) (h : (2 · (Q - P))
>   + (R - Q) = 0) : dist P Q ^ 2 ≠ dist P R ^ 2 - 15 := by sorry
> ```
>
> Natural Language Question:
> ```
> Consider three points `P`, `Q`, and `R` in a plane. If the vector from `Q` to `R`
>   added to twice the vector from `P` to `Q` results in the zero vector, prove
>   that the squared distance from `P` to `Q` is not equal to the squared
>   distance from `P` to `R` minus 15.
> ```

> **Evolved Statement and its corresponding Natural Language Question (Integral)**
>
> Evolved Formal Statement:
> ```
> theorem integral_variant_1
>   (f g : ℝ → ℝ)
>   (hf : ContinuousOn f (Set.Icc 0 1))
>   (hg : ContinuousOn g (Set.Icc 0 1))
>   (h_f_nonneg : ∀ x ∈ Set.Icc 0 1, f x ≥ 0)
>   (h_f_not_zero : ∃ x ∈ Set.Icc 0 1, f x ≠ 0)
>   (h_1 : ∀ x ∈ Set.Icc 0 1, 2 * f x + g x = 0) :
>   ∫ x in (0)..1, (f x)^3 ≠ ∫ x in (0)..1, (g x)^3 - 15 := by sorry
> ```
>
> Natural Language Question:
> ```
> Prove that for any two real-valued continuous functions, `f` and `g`, defined on
>   the interval `[0, 1]`, if `f` is non-negative and not identically zero, and if
>   `2 * f(x) + g(x) = 0` for all `x` in `[0, 1]`, then the integral of `f(x)^3`
>   over `[0, 1]` is not equal to the integral of `g(x)^3` over `[0, 1]` minus 15.
> ```

### A.4 DETAILS FOR EVOLDIFFICULTY

To generate problems of varying difficulty, we define a set of evolution strategies through deliberation and screening by a team of mathematics experts. These strategies fall into two categories:

upward (to increase difficulty) and downward (to decrease difficulty), each with specific methods for implementation.

### A.4.1 UPWARDEVOLUTION

The prompt tempalte for upward evolution is as follows:

---

**Prompt Template for EvolDifficulty (Increase Difficulty)**

Your task is to evolve a given formal statement into several, more complex formal statements, according to the provided strategies and requirements. For each new formal statement, you must provide its corresponding natural language meaning.
### Difficulty Enhancement Strategy

Your objective is to {strategy} for the original statement.

First, understand the core concept and structure of the original formal statement. Identify its key logical components, such as variables, propositions, logical operators, quantifiers, conditions, and the overall scope. Then, you can select from a range of strategies, including but not limited to the following, to enhance difficulty:
{Specific Methods}
...
### Evolution Requirements
Evolved variants should follow the following requirements:
1. Each variant must represent a genuine enhancement of its proof's logic and difficulty, not just an increase in superficial complexity.
2. Each variant should be mathematically rigorous and meaningful
3. Each variant should be syntax correct and a valid Lean 4 statement
4. Each variant should be different from the original statement and other variants
5. Each variant should follow the same output format as "### Example Variant Format".

### Example Variant Format:
```NL Description
Prove that ...
```

```Formal Statement
... := by sorry
```

Please provide 3-5 variants following the strategies and requirements.

### Original Formal Statement
```lean4
{Original Formal Statement}
```

### Evolution Results (3-5 variants with increasing difficulty)

---

The strategies and specific methods are as follows:

---

**Strategies and Specific Methods (Increase Difficulty)**

1. Complicate the Logical Structure
(1) Construct a new problem that increases the nesting depth and layers of the original problem's propositional logic.
(2) Construct a new problem by introducing a logical system with complex dependencies between its components.
(3) Construct a new problem whose internal structure is obscured by multiple layers of

---

non-obvious equivalent transformations.

2. Increase the Mathematical Depth
(1) Construct a new problem that relies on a deeper theoretical framework.
(2) Construct a new problem that requires a longer, but logically similar, chain of reasoning to solve.
(3) Construct a new problem that positions the original problem as a critical sub-problem or lemma within its proof.

3. Elevate Abstraction and Generalization
(1) Construct a new problem by elevating and generalizing a specific instance or special case from the original problem into a universal proposition that must be proven.
(2) Construct a new problem that adds stricter conditions, requiring reasoning and verification under them.
(3) Construct a new problem whose proof requires the fusion of concepts or tools from different knowledge domains.

4. Intensify Constraints and Precision
(1) Construct a new problem that increases complexity by establishing critical boundaries or singularities within the problem's domain.
(2) Construct a new problem that adds specific, strong constraints, requiring the discovery of an optimal solution or an extremal state.
(3) Construct a new problem with heightened rigor requirements, making it necessary to provide a strict argument for the existence, uniqueness, or enumeration of the solution(s).

5. Add Parametric and Analytical Complexity
(1) Construct a new problem that broadens the hypothesis space and increases analytical complexity by introducing or adjusting explicit parameters.
(2) Construct a new problem whose internal structure spans both discrete and continuous forms, requiring a transformation between them (e.g., the limit relationship between a sum and an integral) to be solved.

### A.4.2 DOWNWARDEVOLUTION

The prompt template for downward evolution is as follows:

---

**Prompt Template for EvolDifficulty (Decrease Difficulty)**

Your task is to evolve a given formal statement into several, simpler formal statements, according to the provided strategies and requirements. For each new formal statement, you must provide its corresponding natural language meaning.
### Difficulty Reduction Strategy

Your objective is to {strategy} for the original statement.

First, understand the core concept and structure of the original formal statement. Identify its key logical components, such as variables, propositions, logical operators, quantifiers, conditions, and the overall scope. Then, you can select from a range of strategies, including but not limited to the following, to reduce difficulty:
{Specific Methods}
...
### Evolution Requirements
Evolved variants should follow the following requirements:
1. Each variant must represent a genuine simplification of its proof's logic and structure, not just a cosmetic rephrasing.
2. Each variant should be mathematically rigorous and meanigful
3. Each variant should be syntax correct and a valid Lean 4 statement

---

4. Each variant should be different from the original statement and other variants
5. Each variant should follow the same output format as "### Example Variant Format".

### Example Variant Format:
```NL Description
Prove that ...
```
```Formal Statement
... := by sorry
```

Please provide 3-5 variants following the strategies and requirements.

### Original Formal Statement
```lean4
{Original Formal Statement}
```

### Evolution Results (3-5 variants with decreasing difficulty)

The strategies and specific methods are as follows:

**Strategies and Specific Methods (Decrease Difficulty)**

1. Simplify the Logical Structure
(1) Construct a new problem that decreases the nesting depth and layers of the proposition's logic.
(2) Construct a new problem containing a logical system with weaker or no dependencies between its components.
(3) Construct a new problem whose internal structure is transparent, solvable through direct logical relations rather than non-obvious transformations.

2. Reduce the Mathematical Depth
(1) Construct a new problem that relies on a more elementary theoretical framework.
(2) Construct a new problem that only requires completing the initial steps or the final conclusion of the original problem's longer reasoning chain.
(3) Construct a new problem by isolating a key lemma or an intermediate step from the original problem's proof and setting it as the sole objective.

3. Reduce Abstraction and Specialize
(1) Construct a new problem by taking a general or abstract proposition and creating a specific, concrete instance of it to be solved or verified.
(2) Construct a new problem that replaces abstract symbols and variables with concrete numerical values or tangible examples to lower the barrier to understanding.
(3) Construct a new problem by reformulating it so that it can be solved using concepts and tools from a single, self-contained knowledge domain, avoiding interdisciplinary fusion.

4. Loosen Constraints and Precision
(1) Construct a new problem by restricting its domain to regular cases, excluding critical boundaries or singularities.
(2) Construct a new problem that requires finding any feasible solution rather than an optimal or extremal one.
(3) Construct a new problem that asks for a single concrete example of a solution, rather than a rigorous proof of its existence, uniqueness, or enumeration.

5. Reduce Parametric and Analytical Complexity
(1) Construct a new problem that reduces the dimension of analysis by reducing the

number of variables required to address the problem or by simplifying a complex functional relationship between parameters to a linear one.

### A.4.3 CASE STUDY

We select two examples of EvolDifficulty from evolved dataset. The original Lean 4 statement for upward evolution is as follows:

---

**Original Lean 4 Statement for Upward Evolution**

```
theorem lean_workbook_37427 (n : ℕ) : 133 | 11^(n+2) + 12^(2*n+1) := by sorry
```

---

The evolved Lean 4 statement and its corresponding natural language description are as follows:

---

**Evolved Statement and its corresponding Natural Language Question (Increase Difficulty)**

Evolved Formal Statement:

```
theorem variant_1 (n : ℕ) : 133 | Nat.gcd (11^(n+2) + 12^(2*n+1)) (11^((n+1)+2) +
↪   12^(2*(n+1)+1)) := by sorry
```

Natural Language Question:

```
Prove that for any natural number n, the greatest common divisor of 11^(n+2) +
↪   12^(2*n+1) and 11^((n+1)+2) + 12^(2*(n+1)+1) is divisible by 133.
```

---

---

**Evolved Statement and its corresponding Natural Language Question (Increase Difficulty)**

Evolved Formal Statement:

```
def u_seq : ℕ → ℤ
  | 0 => 133
  | 1 => 3059
  | k + 2 => 155 * u_seq (k + 1) - 1584 * u_seq k
theorem variant_2 n : ℕ : 133 | u_seq n := by sorry
```

Natural Language Question:

```
Let an integer sequence u be defined by u_0 = 133, u_1 = 3059, and the recurrence
↪   relation u_{n+2} = 155 * u_{n+1} - 1584 * u_n for all n ≥ 0. Prove that every
↪   term u_n of the sequence is divisible by 133.
```

---

The original Lean 4 statement for downward evolution is as follows:

---

**Original Lean 4 Statement for Downward Evolution**

```
theorem lean_workbook_21498 (a b : ℝ) (ha : 0 < a) (hb : 0 < b) : (a + b) ^ 2 / 2
↪   + (a + b) / 4 ≥ a * Real.sqrt b + b * Real.sqrt a := by sorry
```

---

The evolved Lean 4 statement and its corresponding natural language description are as follows:

---

**Evolved Statement and its corresponding Natural Language Question (Decrease Difficulty)**

Evolved Formal Statement:

```lean4
theorem variant_1 (a, b : ℝ) (ha : 0 ≤ a) (hb : 0 ≤ b) : (Real.sqrt a + Real.sqrt
↪  b) ^ 2 ≤ 2 * (a + b) := by sorry
```

Natural Language Question:

```
Prove that for any non-negative real numbers `a` and `b`, the square of the sum of
↪  their square roots is less than or equal to twice their sum.
```

---

**Evolved Statement and its corresponding Natural Language Question (Decrease Difficulty)**

Evolved Formal Statement:

```lean4
theorem variant_2 (a, b : ℝ) (ha : 0 < a) (hb : 0 < b) : (a + b) / 2 ≥ Real.sqrt (a
↪  * b) := by sorry
```

Natural Language Question:

```
Prove that Arithmetic Mean-Geometric Mean (AM-GM) inequality for two positive
↪  real numbers `a` and `b`, which states that their arithmetic mean is greater
↪  than or equal to their geometric mean.
```

---

### A.5 DETAILS FOR VERIFICATION

The prompt template to judge consistency between natural language problem and formal statement is as follows:

---

**Prompt Template to Judge Consistency**

You will be provided with a Natural Language Description and a Formal Statement. Please judge if they are consistent, and provide specific analysis:

Natural Language Description:
```Problem
{Natural Language Description}
```

Formal Statement:
```lean4
{Formal Statement}
```

For your response, please follow this example format:
**Consistency Analysis:**
```analysis
Your detailed analysis
```

**Judge Result:**
```judge
Consistent or Inconsistent
```

Now, please provide your formal answer:

---

The prompt template to judge mathematical correctness of formal statements and natural language problem is as follows:

---

**Prompt Template to Judge Correctness**

You will be provided with a Natural Language Description and a Formal Statement. Please judge if the mathematical statement is correct, and provide specific analysis:

Natural Language Description:
```Problem
{original nl}
```

Formal Statement:
```lean4
{correct formal statement}
```

Please analyze the mathematical correctness by considering:
1. Whether the problem is provable (can be proven or disproven)
2. Whether the problem statement is well-formed and meaningful
3. Whether there are any logical contradictions or inconsistencies

For your response, please follow this example format:
**Mathematical Correctness Analysis:**
```analysis
Your detailed analysis
```
**Judge Result:**
```judge
Correct or Incorrect
```

Now, please provide your formal answer:

---

The prompt template for filtering out low-difficulty problems is as follows:

---

**Prompt Template for Filtering Out Low-difficulty Problems**

You will be provided with a Natural Language Description and a Formal Statement. Your task is to classify the difficulty of problem in Lean 4:

Natural Language Description:
```Problem
{Natural Language Description}
```

Formal Statement:
```lean4
{Formal Statement}
```

Please analyze the problem and determine if it is Low-difficulty. Here are the criteria for a Low-difficulty problem:
1. Simple calculations
2. Simple algebraic manipulations
3. Solving single variable linear equations (by just a 1-step calculation)
4. Inequalities proved by an easy sum-of-squares technique

---

Conversely, the following types of problems should NOT be classified as Low-difficulty:
1. Inequality proving with the square root (might be more complex)
2. More complex inequalities, limits, and integrals
3. Problems dealing with integers (more related to number theory)
4. Problems involving higher order roots, complex numbers, matrices, polynomials, group, finite-sum, or functional equations (since these might shed lights on other hard problems)

For your response, please follow this example format:
**Difficulty Analysis:**
```analysis
Your detailed analysis
```
**Judge Result:**
- Is Low-difficulty:
```judge
Yes or No
```

Now, please provide your formal answer:

Prompt template for fixing compilation errors in a formal statement is as follows:

### Prompt Template for Correcting Formal Statement

Your task is to fix the code based on the errors and provide a corrected version. Please also provide a detailed analysis of the changes you made. You will be provided with an incorrect Lean4 code snippet and a list of corresponding errors.

Incorrect Lean4 Code:
```lean4
incorrect lean4 code
```
Error List:
```errors
errors
```

Please modify the incorrect Lean 4 code according to the following requirements:
1.The corrected statement must be syntactically valid and well-typed according to Lean4 rules.
2.The correction should maintain the original mathematical meaning that the user was likely trying to express in the statement.
3.The corrected Lean 4 code must end with ':= by sorry'.

For your response, please follow this example format:
**Modification Analysis**
```analysis
Your detailed analysis
```
**Corrected Lean4 Code**
```lean4
Your corrected Lean4 code
```

Now, please provide your formal answer:

## A.6   CASE STUDY FOR EVOLAST

EvolAST is an approach that integrates code features(Tian et al., 2025; Yan et al., 2023; Luo et al., 2026) with mathematical reasoning, making it highly suitable for formal reasoning tasks in Lean4 code generation. We select an example of EvolAST from evolved dataset. The original Lean 4 statement is as follows:

---

**Original Lean 4 Statement**

```
theorem evolved_thm (x, y : ℝ) (h_0 : x * y = 4) (h_1 : x > y) (h_2 : x^3 - y^3 =
↪  3555) : x^2 + y^2 = 233 := by sorry
```

---

The evolved Lean 4 statement is as follows:

---

**Evolved Statement**

Evolved Formal Statement:

```
theorem evolved_thm_auged (x, y : ℝ) (h_1 : y < x) (h_2 : 3555 = x^3 - y^3) (h_0 : 4
↪  = y * x) :  233 = y^2 + x^2 := by sorry
```

---

## A.7   DETAILS OF SIMILARITY EVALUATION

We report here the full prompt template used for the LLM-based similarity assessment.

---

**Prompt Template for Similarity Evaluation**

Your task is to evaluate whether the two given formal math statements are similar, not just looking similar or involving overlapping mathematical concepts. Please analyze both statements carefully, focusing on:
1. The overall structure of the statements (e.g., setup, sequence of steps, logical flow)
2. The specific mathematical operations or reasoning paths required,
3. The wording and presentation style,
4. Whether one statement appears to be a trivial rephrasing or numerical variant of the other.

Provide a similarity score from 1 to 10, where:
1 = completely different statements,
10 = semantically identical statements.

Here are provided statements:
Statement1:
{statement1}
Statement2:
{statement2}
Format your response as:
```
<reason>Reason for your score</reason>
<score>Your score(1-10)</score>
```

---

## A.8   ANALYSIS OF OUTPUT LENGTH

Compared to other non-reasoning models, EvolProver exhibits a higher average token consumption, primarily due to the design of its instruction-tuning stage. In this stage, the model is trained to repeat the problem statement (including both the formal statement and its corresponding natural language semantic annotation) before generating the proof. In contrast, some other non-COT models, such as DeepSeek-Prover-V1.5 and Goedel-Prover-SFT, are trained to directly produce the proof. Consequently, a non-reasoning segment (the repeated problem statement) is included in EvolProver's outputs and is counted in its token usage.

| Benchmark | Average Token Length |
|-----------|---------------------|
| FormalMATH-lite | 407.3 |
| Minif2f | 440.8 |

Table 6: Average proof-only token length of EvolProver after removing the problem restatement.

| Model | Average Token Length | Tactic Steps |
|-------|---------------------|--------------|
| EvolProver-Base | 113.75 | 9 |
| EvolProver | 129.05 | 10 |

Table 7: Average token length and tactic steps of EvolProver and EvolProver-Base on the subset of commonly solved problems.

To quantify this effect, we additionally compute the token length for the proof-only portion of EvolProver after removing the problem restatement. The results are shown in Table 6.

We believe that the token length of the output containing only the proof is on the same scale as that of other non-COT models. Additionally, we measured the output length and the number of executed tactic steps on the subset of problems solved by both EvolProver and its baseline. We consider tactic steps to be a good representation of the verified proof length to some extent. The results are shown in Table 7.

As we can see, EvolProver has slightly higher values in both Average Token Length and tactic steps compared to EvolProver-Base. We carefully examined their proofs and selected a representative example for comparison:

---

**Representative Lean 4 proofs of EvolProver-Base and EvolProver**

```
-- Proof of EvolProver-Base
theorem example_base {x : } (hx : x  0) (h : 1 / 9 + 1 / 18 = 1 / x) :
    x = 6 := by
  field_simp at h
  linarith

-- Proof of EvolProver
theorem example_evolved {x : } (hx : x  0) (h : 1 / 9 + 1 / 18 = 1 / x) :
    x = 6 := by
  have h : x = 6 := by
    have h : 1 / x = 1 / 9 + 1 / 18 := by linarith
    have h : 1 / x = 1 / 6 := by norm_num at h
    have h : x = 6 := by
      have h : x  0 := hx
      field_simp at h
      nlinarith
    exact h
  exact h
```

---

We found that the response from EvolProver-Base is more intuitive, with a direct approach and lower readability, while EvolProver's response is more reasoning-based, habitually showing the conditions and providing a clear proof. However, this results in a longer output.

## A.9  USE OF LARGE LANGUAGE MODELS

We utilized Large Language Models (LLMs) solely to refine the language and improve the clarity of this manuscript.

