# OpenReview forum: "EvolProver: Advancing Automated theorem proving by Evolving Formalized Problems via Symmetry and Difficulty"
_ICLR.cc/2026/Conference — ICLR 2026 Poster_

### Official Review · Reviewer_fmUh · 2025-10-29

**Soundness:** 2
**Presentation:** 3
**Contribution:** 2
**Rating:** 4
**Confidence:** 4

**Summary:**

This paper presents EvolProver, a 7B-parameter non-reasoning theorem prover enhanced through a novel data augmentation pipeline that addresses both symmetry and difficulty aspects of formalized mathematical problems. The pipeline consists of three complementary methods: EvolAST for syntactic symmetry transformation, EvolDomain for cross-domain semantic symmetry, and EvolDifficulty for generating problems with varying difficulty levels. Experimental results demonstrate that EvolProver achieves good performance on multiple benchmarks, including FormalMATH-Lite (53.8% pass@32), MiniF2F-Test (69.8% pass@32), and Ineq-Comp datasets, outperforming comparable non-reasoning models and even some reasoning-based models while using significantly fewer tokens.

**Strengths:**

1. The proposed data augmentation pipeline demonstrates a novel and systematic approach to improving model generalizability by addressing both syntactic and semantic symmetry, as well as difficulty variation, which is a comprehensive solution to the fragility issue in formal theorem proving.​
2. The ablation studies and controlled experiments effectively validate the contribution of each component in the pipeline, showing consistent performance improvements across multiple benchmarks and training stages.

**Weaknesses:**

1. The performance advantage of EvolProver over DeepSeek-Prover-V2-7B-non-CoT on MiniF2F-Test is relatively modest (69.8% vs 68.0±0.5%), while DeepSeek-Prover-V2-7B-non-CoT uses fewer average tokens, raising questions about the practical significance of the improvement.​
2. The case study in Figure 1 does not clearly demonstrate how the augmented dataset helps solve problems that were previously unsolvable, making it difficult to assess the qualitative impact of the data augmentation.​
3. The ablation experiments do not include a detailed analysis of how each component contributes to performance on different mathematical domains, which would help understand the specific strengths of each augmentation method.

**Questions:**

1. Why wasn't DeepSeek-Prover-V2-7B-non-CoT included in the comparisons for FormalMATH and Ineq-Comp benchmarks, given its strong performance on MiniF2F-Test? Including these comparisons would provide a more comprehensive evaluation of EvolProver's capabilities across different benchmarks.​
2. The example shown in Figure 1 appears to involve relatively minor tactic changes in Lean that may not significantly affect the proof difficulty. Could you provide more case studies demonstrating specific instances where the augmented dataset enabled EvolProver to solve problems that the baseline model (EvolProver-Base) or other state-of-the-art models could not solve?

---

> ### Author Response · Authors · 2025-11-22
> **Response to Reviewer fmUh (1/4)**
>
> Dear Reviewer fmUh,
>
> Thank you for your detailed feedback and for raising several important points for discussion.
>
> ---
>
> **Response to Weakness #1: The performance advantage of EvolProver over DeepSeek-Prover-V2-7B-non-CoT on MiniF2F-Test:**
>
> Regarding the performance on MiniF2F-Test, we would like to offer some points of context to clarify the contribution of our work:
>
> a.  **Broader Benchmark Performance:** Not only the performance on MiniF2F-Test is higher, EvolProver demonstrates a much more significant advantage on the larger and more comprehensive **FormalMATH benchmark**. As shown in our results, we outperform not only the non-CoT version but also the reasoning-based DeepSeek-Prover-V2-7B, establishing a new SOTA. This indicates superior generalization ability.
>
> b.  **Training without CoT Data:** We have reason to believe that the DeepSeek-Prover-V2-7B non-CoT model shares weights or training data with its CoT counterpart, meaning it was implicitly trained on CoT data.  In contrast, EvolProver was trained **without any CoT data whatsoever**, which greatly reduces computational cost. Achieving SOTA performance under these more challenging and computationally less expensive conditions highlights the remarkable effectiveness of our data augmentation strategy.
>
> c.  **A novel data synthesis pipeline:** We propose a formal-to-formal data synthesis pipeline that differs from existing work. In this pipeline, EvolDomain and EvolDifficulty are used to enhance the diversity of the data in terms of domain and difficulty, respectively, while EvolAST is a formal mathematical data synthesis method that does not rely on LLMs and combines code structure with mathematical semantics. Experimental results show that our method not only achieves strong performance, but is also more efficient than current mainstream informal-to-formal data synthesis pipelines.
>
> d. **Response Pattern**: Compared to other non-reasoning models, EvolProver's higher average token consumption is a result of our design during the instruction-tuning stage. In this stage, **the model learns to repeat the problem statement (including the statement and its corresponding natural language annotation of the semantics) before providing the proof.** In contrast, some other non-COT models, such as DeepSeek-Prover-V1.5 and Goedel-Prover-SFT, directly generate the proof section. While this portion of token consumption is not part of the reasoning process, it does appear in our output.
>
> To further clarify, we have computed the token length for the proof-only portion of EvolProver and DeepSeek-Prover-V2-7B-non-CoT after removing the problem restatement.
>
> EvolProver:
>
>     | Benchmark       | Average Token Length |
>     | --------------- | :------------------: |
>     | FormalMATH-lite |        407.3         |
>     | Minif2f         |        440.8         |
>
>
> DeepSeek-Prover-V2-7B-non-CoT:
>
>     | Benchmark       | Average Token Length |
>     | --------------- | :------------------: |
>     | FormalMATH-lite |        413.9         |
>     | Minif2f         |        457.2         |
>
> We believe that the token length of the output containing only the proof is on the **same scale** as that of **DeepSeek-Prover-V2-7B-non-CoT**.
>
> ---
>
>
> **Response to Weakness #2: clarity of the case study in Figure 1:**
>
> We understand your concern about Figure 1. The primary goal of Figure 1 is to **illustrate the mechanics of our augmentation pipeline**—showing *how* a theorem is transformed at each stage—rather than to provide a direct causal link for how one augmented example enables solving a new problem. The example was specifically chosen for its clarity and ease of understanding. We provide multiple, more complex case studies in the Appendix 3,4,6.
>
> ---

---

> ### Author Response · Authors · 2025-11-22
> **Response to Reviewer fmUh (2/4)**
>
> **Response to Weakness #3. Regarding the ablation study:**
>
>
> Thank you for this excellent suggestion. Understanding how each augmentation component impacts performance across different mathematical domains is indeed crucial for a deeper analysis. We have conducted this detailed breakdown, and the results are summarized in the table below. This analysis reveals the specific strengths of each method and will be included in the final version of our paper.
>
>
>
>
> | Domain                 | EvolProver-Base | EvolProver-Ablation-SFT | EvolProver-SFT | EvolProver-Ablation-RL | EvolProver |
> |------------------------|-----------------|-------------------------|----------------|------------------------|------------|
> | Algebra                | 121             | 133                     | 134            | 133                    | 141        |
> | Applied Mathematics    | 28              | 30                      | 32             | 32                     | 33         |
> | Number Theory          | 16              | 21                      | 23             | 23                     | 23         |
> | Precalculus            | 14              | 15                      | 15             | 15                     | 15         |
> | Geometry               | 7               | 8                       | 8              | 8                      | 8          |
> | Discrete Mathematics   | 2               | 3                       | 3              | 5                      | 5          |
> | Calculus               | 0               | 2                       | 2              | 3                      | 3          |
> | Multivariable Calculus | 2               | 2                       | 2              | 2                      | 2          |
> | Others                 | 0               | 0                       | 0              | 0                      | 0          |
>
>
> - For EvolProver-Ablation-SFT (retaining only EvolDomain and EvolDifficulty), compared with Base, almost all domains show improvements to varying degrees, but the two augmentations operate in different regimes: in long-tail domains where samples are relatively scarce, the gains mainly come from EvolDomain. By performing cross-domain semantic evolution, it transfers existing structural knowledge from high-resource domains, thereby to some extent compensating for the gaps in data coverage in these low-resource domains; whereas in high-resource domains where samples are relatively sufficient and the model has basically mastered the core patterns, EvolDifficulty plays the primary role. By systematically widening the difficulty distribution of problems, it enables the model to acquire stronger problem-handling ability on top of its original capability. Overall, EvolDomain mainly alleviates data scarcity in low-resource domains, while EvolDifficulty significantly enhances the model’s robustness to problems of different difficulty levels in high-resource domains, and the two are complementary across domains with different data scales.
>
> - EvolProver-SFT is the SFT result obtained by further adding EvolAST on top of EvolProver-Ablation-SFT. Compared with Ablation-SFT that only uses EvolDomain and EvolDifficulty, EvolProver-SFT achieves small but consistent additional improvements in domains with a higher proportion of symbolic reasoning such as algebra, number theory, and applied mathematics (e.g., Algebra: 133 to 134, Applied Mathematics: 30 to 32, Number Theory: 21 to 23), while performance in other domains remains basically unchanged. This indicates that EvolAST does not significantly introduce new semantic knowledge, but by generating multiple equivalent structured representations around the same reasoning process, it strengthens the model’s ability to handle different forms with the same meaning, thereby bringing extra generalization gains in high-resource domains that already have a certain sample size and semantic diversity.
>
> - EvolProver-Ablation-RL is obtained by further performing RL optimization on top of EvolProver-Ablation-SFT using data generated by EvolDomain and EvolDifficulty. As can be seen from the table, compared with EvolProver-SFT, performance in high-resource domains (such as Algebra, Applied Mathematics, Number Theory, etc.) remains overall stable, while additional gains appear in sparser, more difficult, or larger-search-space domains (e.g., Discrete Mathematics: 3 to 5, Calculus: 2 to 3). This shows that under a data distribution that combines cross-domain coverage and difficulty evolution, RL can make fuller use of these evolved samples through exploratory training.

---

> ### Author Response · Authors · 2025-11-22
> **Response to Reviewer fmUh (3/4)**
>
> - The final EvolProver performs RL on the full dataset that includes all three augmentations (EvolAST, EvolDomain, and EvolDifficulty). Compared with EvolProver-Ablation-RL, it achieves further improvements in high-resource domains such as Algebra (133 to 141) and Applied Mathematics (32 to 33), while maintaining the gains in Discrete Mathematics (5) and Calculus (3). Overall, this reflects the complementary roles of the three augmentations along the three dimensions of domain coverage, difficulty distribution, and structural diversity.
>
>
>
> This analysis reveals the specific strengths of each method and will be included in the final version of our paper.
>
> Thank you for helping us strengthen our paper with your valuable comments.
>
> ---
>
> **Response to Question #1: the performance comparison with DeepSeek-Prover-V2-7B-non-CoT**
>
> It is generally observed that within the same model family, the non-CoT mode typically **underperforms** compared to the CoT mode. Consequently, previous works introducing new benchmarks, such as FormalMATH and Ineq-Comp, usually report only the best results achieved by DeepSeek-Prover-V2-CoT.
>
> In our comparative experiments, we adhered to the standard practice of citing results directly from the original papers to ensure a fair comparison, and thus did not initially run additional tests for settings not reported by the authors. However, we are happy to provide the performance of DeepSeek-Prover-V2-7B-non-CoT for a more comprehensive evaluation. The specific test results are as follows:
>
> | Model                         | FormalMATH-Lite | Ineq-Comp (Seed) | Ineq-Comp (Transformed) | Ineq-Comp (Ratio) |
> | ----------------------------- | ---------: | ---------------: | ----------------------: | ----------------: |
> | DeepSeek-Prover-V2-7B-non-CoT |     50.35% |           56.00% |                  27.33% |            48.90% |
> | EvolProver                    |     53.86% |           52.20% |                  34.02% |            65.17% |
>
> ---
>
> **Response to Question #2:  specific instances where the augmented dataset enabled EvolProver to solve problems:**
>
> We respecfully point out that, generally speaking, performance improvements arise from the distributional shifts of **the entire training set**, and attributing a new capability to **a single data point is technically infeasible**. To better showcase the qualitative impact, we provide a qualitative analysis below, where we present a concrete example of a theorem that was unsolvable by the baseline model but are successfully proven by EvolProver. The proof below was generated by EvolProver.
>
>
> ```lean
> import Mathlib
> open Real Set
> open scoped BigOperators
> theorem algebra_569189 (x : ℝ) :
>   |sqrt (x^2 + x + 1) - sqrt (x^2 - x + 1)| < 1 := by
>   have h1 : sqrt (x^2 + x + 1) + sqrt (x^2 - x + 1) > 0 := by
>     have h1 : sqrt (x^2 - x + 1) ≥ 0 := by exact sqrt_nonneg (x ^ 2 - x + 1)
>     have h2 : sqrt (x^2 + x + 1) > 0 := by
>       have h1 : (x ^ 2 + x + 1 : ℝ) > 0 := by
>         nlinarith [sq_nonneg (x + 1/2)]
>       exact sqrt_pos.mpr h1
>     linarith
>   have h2 : (sqrt (x^2 + x + 1) - sqrt (x^2 - x + 1)) ^ 2 < 1 := by
>     have h1 : sqrt (x^2 + x + 1) * sqrt (x^2 - x + 1) ≥ 0 := by
>       apply mul_nonneg
>       · exact sqrt_nonneg (x ^ 2 + x + 1)
>       · exact sqrt_nonneg (x ^ 2 - x + 1)
>     nlinarith [sq_sqrt (show (x ^ 2 + x + 1 : ℝ) ≥ 0 by nlinarith [sq_nonneg (x + 1/2)]),
>       sq_sqrt (show (x ^ 2 - x + 1 : ℝ) ≥ 0 by nlinarith [sq_nonneg (x - 1/2)]),
>       sq_nonneg (sqrt (x^2 + x + 1) - sqrt (x^2 - x + 1)),
>       sq_nonneg (sqrt (x^2 + x + 1) + sqrt (x^2 - x + 1)),
>       sq_nonneg (sqrt (x^2 + x + 1) - sqrt (x^2 - x + 1)),
>       h1]
>   have h3 : -1 < sqrt (x^2 + x + 1) - sqrt (x^2 - x + 1) := by
>     nlinarith [sq_nonneg (sqrt (x^2 + x + 1) - sqrt (x^2 - x + 1)), h2]
>   have h4 : sqrt (x^2 + x + 1) - sqrt (x^2 - x + 1) < 1 := by
>     nlinarith [sq_nonneg (sqrt (x^2 + x + 1) - sqrt (x^2 - x + 1)), h2]
>   have h5 : -1 < sqrt (x^2 + x + 1) - sqrt (x^2 - x + 1) ∧ sqrt (x^2 + x + 1) - sqrt (x^2 - x + 1) < 1 := by
>     constructor
>     · exact h3
>     · exact h4
>   have h6 : sqrt (x^2 + x + 1) - sqrt (x^2 - x + 1) < 1 := by exact h4
>   have h7 : -1 < sqrt (x^2 + x + 1) - sqrt (x^2 - x + 1) := by exact h3
>   rw [abs_lt]
>   constructor
>   · nlinarith
>   · nlinarith
> ```
>
> ---

---

> ### Author Response · Authors · 2025-11-22
> **Response to Reviewer fmUh (4/4)**
>
> Thank you for your time and valuable feedback. We believe our detailed response addresses the concerns raised in your review. If any points remain unclear, we are eager to answer further questions. If our response has alleviated your concerns, we respectfully request that you reconsider your evaluation of our paper.

---

### Official Review · Reviewer_RmgK · 2025-10-30

**Soundness:** 2
**Presentation:** 3
**Contribution:** 3
**Rating:** 4
**Confidence:** 4

**Summary:**

This paper proposes three data augmentation methods for formal mathematics statements based on symmetry and difficulty. EvolAST: apply seven prewritten transformation rules at random on the theorem’s AST to produce logically equivalent variants; EvolDomain: translate problems across different domains while preserving their core logic, guided by evolutionary instructions for the LLM to perform the conversions; EvolDifficulty: use an LLM with instructions to adjust the difficulty of existing theorems and generate statements at varying difficulty levels. Using this pipeline to augment public datasets (e.g., STP and DeepSeek Prover V1) and training on DeepSeek Prover V1.5 base, the authors build EvolProver. Experiments show strong performance the proposed model among non-CoT 7B ATP models.

**Strengths:**

1. The proposed EvolAST method is novel: it leverages Lean 4 features in an intuitive and effective way. The results in Table 2 on the Ineq-Comp dataset show that this augmentation can substantially mitigate non-CoT provers’ fragility to the specific form of a statement.

2. From Figure 5, the EvolDomain approach appears to significantly alleviate the concentration of formal-statement datasets in algebra and number theory. This may be valuable for improving ATP generalization across different mathematical domains.

3. The ablation study is thorough. Table 4 presents multiple combinations of SFT, RL, and different augmentation methods, providing useful insight into which components are effective. The comparison between directly augmenting formal data and the more traditional approach of augmenting natural-language data first and then translating it is also a useful addition.

4. Figures and tables are clear and easy to understand, and the paper is well written overall.

**Weaknesses:**

1. Choice of an old base model and non-CoT approach, which appears unnecessary and hard to justify. The paper trains EvolProver on a non-CoT DeepSeek V1.5 7B model. The three proposed data-augmentation methods should in principle be directly applicable to CoT-style models (e.g., DeepSeek Prover v2 or Goedel Prover v2), so restricting experiments to a non-CoT, relatively old base model limits the relevance of the work given that much of the recent ATP community has shifted to CoT-capable reasoning models. EvolProver’s performance is still substantially weaker than contemporary reasoning models. The authors should further explain why they chose a non‑CoT model and an older base model.
2. Small amount of synthesized data and no cost analysis. The amount of augmented/synthesized data is relatively small and the paper does not discuss the cost of data synthesis. Line 653 states that the pool of publicly available datasets used is 3.3 million examples, but only 39.2k examples were synthesized (Line 665). The authors do not explain the roughly two-order-of-magnitude gap (e.g., due to cost or filtering?), nor do they provide analysis of synthesis cost, token usage, or API expenses.
3. Missing proportions of original vs. augmented data in SFT and RL. The paper does not report the ratio of original to augmented examples during supervised fine-tuning (SFT) and reinforcement learning (RL), which hurts reproducibility. Given the large disparity in size between the original and augmented datasets, the mixing ratio is an important experimental detail and should be provided.
4. Minor writing and presentation issues (do not affect readability but should be fixed to improve manuscript quality)

- Line 124: missing space before a citation.
- Lines 143, 153, 227: inconsistent capitalization of “Lean4” vs. “Lean 4”; terms should be made consistent.
- Table 4 and Table 5 appear to contain the same content and may be duplicated.
- Related work: the survey of data-augmentation methods focuses on informal mathematics; the authors should also cover data-augmentation methods specific to formal mathematics. (eg., STP and Ineq-Comp)

##

**Questions:**

1. Can the three proposed augmentation methods be applied to chain-of-thought reasoning models? What obstacles might arise from directly applying them to CoT-style reasoning models? The paper claims that current provers are fragile to even minor transformations of problem statements — does this fragility persist in contemporary reasoning provers like Deepseek Prover v2 or Geodel Prover v2?

2. Line 142 states that EvolAST is introduced "to mitigate this issue (the inevitable introduction of syntactic or semantic errors)," yet Figure 2 places the EvolAST process after EvolDomain and EvolDifficulty. This ordering seems odd: errors can propagate, so a theorem that becomes semantically incorrect (e.g., unprovable) after EvolDomain or EvolDifficulty would not be fixed by applying EvolAST afterward. Please provide further explanation for this pipeline ordering and clarify how EvolAST mitigates syntactic/semantic errors introduced by earlier stages.

---

> ### Author Response · Authors · 2025-11-22
> **Response to Reviewer RmgK (1/3)**
>
> Dear Reviewer RmgK,
>
> Thank you for your valuable feedback and insightful comments on our work! Below, we provide a comprehensive response to the questions and concerns you raised.
>
> ---
>
> **Response to Weakness #1:  On not fine-tuning from a CoT model and tune on an 'old' model**
>
> As we argue in the introduction, a core motivation of our work is to specifically enhance the capabilities of **non-reasoning provers**. **As argued by reviewer uSHo and observed repeatedly in both industry and academia**(for example, in Hilbert Agent Framework[1] and Grok Code Fast 1[2]), **fast, non-reasoning models are critical for agentic AI**, as the high latency of chain-of-thought models would be prohibitive. Our goal was to investigate the untapped potential of data augmentation for this specific class of models. Fine-tuning from a reasoning model would be an interesting but fundamentally different research direction, misaligned with our stated goal of advancing fast, efficient non-reasoning theorem proving. It is also worth noting that our model achieves performance comparable to, or even surpassing, that of some reasoning provers on several benchmarks.
>
>
> Furthermore, regarding our choice of the base model, we would like to clarify that starting from DeepSeek-Prover-V1.5-Base is **not** a choice of an 'old' model, but rather a standard and methodologically sound practice. In fact, the very models mentioned by the reviewer, DeepSeekProver-V2-7B, are themselves further fine-tuned from this **exact same** base model. By starting from the same foundation, our work provides a clear and direct comparison of the efficacy of different data generation and fine-tuning strategies. It isolates the contribution of our Evol pipeline from the influence of a different, more capable base model, thus providing a fairer and more insightful analysis.
>
> ---
>
> **Response to Weakness #2: Regarding the size gap between augmented data and raw data**
>
> As mentioned in Appendix A.1, our 39.2k examples are synthesized from a sample of size 70k. These initial 70k samples were carefully curated from our four sources. Three of the datasets—`DeepSeek-Prover-V1`, `FormalMATH-All (excluding FormalMATH-Lite)`, and `MiniF2F-Valid`—are high-quality but relatively small, collectively containing just over 30,000 theorems. We included all theorems from these valuable sources in our synthesis pool. The fourth source, `STP-lean`, is a much larger synthetic  dataset, with **3.26 million** theorems. To prevent this synthetic data from dominating the more curated sources and avoid excessive computational costs, we first filtered out theorems originating from mathlib. From the remainder, we performed a weighted sampling to select an additional 30,000+ examples, guided by the importance weights provided by the STP-lean authors for each theorem. The number of sampled examples was chosen to be **comparable to the total size of the other real-world datasets combined**.
>
> This strategy allowed us to create a high-quality and diverse seed set of 70k theorems in a cost-effective manner, **balancing the inclusion of valuable real-world problems with a broader set of synthetic examples**. We will clarify this selection process more explicitly in the main body of the paper to avoid confusion. Thank you for bringing this to our attention.
>
> ---
>
> **Response to Weakness #2: On computational cost:**
>
> We provide a detailed token costs associated with our data augmentation pipeline below:
>
> |           Stage                                           | Token Consumption |
> |----------------------------------------------|:-----------------:|
> | Problem synthesis (EvolDomain & EvolDifficulty) | 893M              |
> | Verification process                           | 506M              |
> | EvolAST                                        | 0                 |
> | Proof synthesis                                | 11B               |
>
> Thank you very much for pointing that out. We will include it in the new version of our paper.
>
> ---

---

> > ### Comment · Reviewer_RmgK · 2025-11-26
> >
> > Thank you for the rebuttal. The clarifications described should improve the presentation and clarity of the paper. I would maintain my current score.

---

> > > ### Author Response · Authors · 2025-11-27
> > >
> > > Thank you again for your valuable feedback.  We have carefully incorporated all of the suggested clarifications into the revised manuscript, which we believe has improved the overall presentation and clarity of the paper.
> > >
> > > Could you please let us know if there are any remaining major concerns?  We would be very happy to provide any further clarifications or revisions as needed.

---

> ### Author Response · Authors · 2025-11-22
> **Response to Reviewer RmgK (2/3)**
>
> **Response to Weakness #3: On the proportions of original vs. augmented data:**
>
> We agree that the description in Appendix A.1 was not sufficiently clear. In the SFT stage, we used all the theorems that come with proofs, including public data (3.3M) and our augmented data (39.2K). It is worth noting that, as detailed in the ablation study in Table 4, the 39.2K augmented data plays a critical role. Despite the vast size of the 3.3M public data baseline, the addition of the augmented data resulted in significant gains, proving its effectiveness.
>
> In the RL stage, we used our augmented data (39.2K) and additionally incorporated FormalMATH-All (excluding FormalMATH-Lite) and MiniF2F-Valid into our training data. We will clearly specify the composition of our training data in the final version of the paper.
>
> ---
>
> **Response to Weakness #4. On Minor writing and presentation issues**
>
> Thank you for pointing this out. We will revise these minor issues in the new version of our paper.
>
> ---
>
> **Response to Question #1: On applying our methods to CoT models and the fragility of reasoning provers:**
>
> This is an excellent question.
>
> *   **Applicability to CoT models:** In principle, our augmentation methods can certainly be used to generate novel problems for training CoT models. The primary obstacle to directly applying them is not conceptual but resource-related—namely. The generation of corresponding chain-of-thought solutions for each newly synthesized problem is computationally intensive and costly. Therefore, while entirely feasible, extending our pipeline to CoT models is primarily a matter of securing sufficient computational resources for this solution-generation phase.
> *   **Fragility of contemporary reasoning provers:** We respectfully point out that experiments in our paper already showed that even SOTA reasoning provers remain fragile to transformations. On Table 2, the performance of contemporary  models, **including DeepSeek-Prover-V2(CoT)**, drops significantly under small transformations, while our prover maintains a stronger footing. This result underscores the broad relevance of our work in improving model robustness, a challenge that persists across both reasoning and non-reasoning architectures.
>
> ---

---

> ### Author Response · Authors · 2025-11-22
> **Response to Reviewer RmgK (3/3)**
>
> **Response to Question #2: On the ordering of the augmentation pipeline (EvolAST after EvolDomain/EvolDifficulty):**
>
> Thank you for this insightful question. The pipeline ordering is a deliberate design choice to mitigate the risk of instability amplification. We conceptualize the potential instability introduced by each stage with a simple model.
> *   `EvolDomain` and `EvolDifficulty`, being LLM-based, can introduce significant and unpredictable variations. The potential for instability propagation can be likened to an **exponential function**, where a small initial instability `x` can become `exp(x)`.
> *   `EvolAST`, being a deterministic, syntax-based transformation, introduces much less instability. Its effect is more akin to a **linear function**, transforming `x` to `2x`.
>
> Given this, applying `EvolAST` first would lead to `x → 2x → exp(2x)`, where the instability is scaled *before* being passed through the exponential, potentially leading to a massive accumulation of instability. Our chosen order, `x → exp(x) → 2exp(x)`, first contains the LLM-induced instability and then applies the more controlled syntactic transformation. This sequence proves to be more stable and effective at generating high-quality, diverse, yet valid theorems.
>
> Controlling instability is crucial because, as noted in our paper and  other studies cited like Ineq-Comp[3], excessive instability can degrade model performance. We are particularly concerned that highly unstable problems may become intractable for proof generation. Such intractability would directly hinder our ability to collect a sufficient volume of valid training data, which is essential for the success of our data augmentation strategy. Therefore, we selected the configuration that yields moderate overall instability.
>
> We will add this explanation to the paper to clarify our methodology. In future work, we plan to systematically explore alternative augmentation sequences to further validate this hypothesis.
>
> ---
>
> *reference*
>
> [1]Varambally, Sumanth, et al. "Hilbert: Recursively Building Formal Proofs with Informal Reasoning."
>
> [2]https://x.ai/news/grok-code-fast-1
>
> [3]Zhao, Geng, et al. "Ineq-Comp: Benchmarking Human-Intuitive Compositional Reasoning in Automated Theorem Proving on Inequalities."
>
> ---
>
> We sincerely appreciate the time and effort you devoted to reviewing our paper and for your constructive comments. It is our hope that our point-by-point responses have cleared up your concerns; if anything is still not fully addressed, please feel free to reach out with additional questions. If these clarifications are helpful, we would humbly and respectfully ask you to consider updating your assessment of our work.

---

### Official Review · Reviewer_7CuZ · 2025-10-31

**Soundness:** 3
**Presentation:** 3
**Contribution:** 3
**Rating:** 6
**Confidence:** 3

**Summary:**

The paper presents a data augmentation pipeline aimed at improving the robustness and generalization of large language models (LLMs) for formal theorem proving. The proposed system combines three key modules:

* EvolAST – syntactic augmentation using Abstract Syntax Tree-based transformations that preserve semantics while changing structure;

* EvolDomain – semantic augmentation by translating theorems across mathematical domains (e.g., geometry ↔ algebra) using LLMs;

* EvolDifficulty – difficulty modulation via evolutionary instructions to generate problems of varying complexity.

After automated verification, the augmented data is used to train EvolProver, a 7B-parameter non-reasoning model that achieves new SOTA performance on multiple formal math benchmarks (e.g., FormalMATH-Lite, MiniF2F, Ineq-Comp). Ablations confirm each augmentation’s benefit. The approach focuses on data diversity rather than architectural changes.

**Strengths:**

1. Practical and well-motivated. They address a real problem on the fragility and poor generalization of theorem-proving LLMs, and the strategy they use (data-centric improvements) is simple but effective, much easier than building a reasoning model in practice.

2. Systematic pipeline design. The decomposition into syntactic (EvolAST), semantic (EvolDomain), and difficulty-based (EvolDifficulty) augmentations is both intuitive and well-structured.

3. Results are good. The model achieves consistent SOTA performance across diverse benchmarks, including non-reasoning categories where size-matched baselines are surpassed. They can achieve even pretty close performance compared to the SOTA reasoning model baselines. Ablations studies convince us into believing that the pipeline really helps, rather than overfitting to artifacts.

**Weaknesses:**

1. The paper remains more empirical than theoretical, as there's no deeper analysis of why the augmentations help, or how they relate to reasoning robustness.

2. There comes concerns on whether or not the framework relies on LLM-generated data too much. EvolDomain and EvolDifficulty both depend on LLM quality, thus risk subtle semantic drift or spurious logic, which might not be fully detected by verification.

3. Only benchmarks with formal Lean-style problems are tested, how robust the framework is when generalizing to other theorem systems, or other symbolic reasoning datasets?

4. The term "evolutionary" may be overstated a little bit?

**Questions:**

Addressing the questions mentioned in the weakness points should be enough.

---

> ### Author Response · Authors · 2025-11-22
> **Response to Reviewer 7CuZ (1/2)**
>
> Dear Reviewer 7CuZ,
>
> We really appreciate your helpful feedback and thoughtful comments on our paper! In what follows, we address each of your questions and concerns one by one.
>
> ---
>
> **Response to Weakness #1: Empirical Focus and Unexplained Augmentation Benefits**
>
> We agree your concerns regarding our focus on experiments. The core focus of this work is to systematically enhance the model’s generalization ability through multiple data augmentation methods. These methods enrich and restructure the training data from different dimensions, significantly improving the model’s adaptability and robustness to various types of problems. Our experimental results show that, by expanding data diversity and the difficulty spectrum, the model’s robustness has been effectively improved, leading to clear performance gains on multiple reasoning benchmarks.
>
> ---
>
> **Response to Weakness #2: Hidden Biases from LLM-Dependent Data Generation in EvolDomain/EvolDifficulty**
>
> In fact, in the domain of formal languages, almost every LLM-dependent data synthesis pipeline suffers from this issue, and one of the **original motivations of our method is to reduce semantic drift.** Many works, such as DeepSeek-Prover, Kimi-Prover, etc., include autoformalization process. However, **relying solely on natural-language math problems followed by an LLM for formalization may not be the most efficientmor easily controllable pipeline.** The main reasons are that:
>
> - it is difficult to completely avoid semantic biases in the data used to train the formalizer, and the formalizer may further amplify such issues during inference;
>
> - this route does not fully exploit those high-quality formalized statements that have already been rigorously proved in theorem-proving systems.
>
> Against this background, we designed our current methods and experimental setup in an attempt to alleviate the above issues to some extent in practice, specifically as follows:
>
> **(1) Evolution based on rigorously proved formal theorems**
> Most of our seed theorems are drawn from existing Lean formalization datasets (e.g., DeepSeek-Prover-V1, STP-Lean, etc.). These theorems already have complete formal proofs in Lean and have been successfully verified by the Lean server; in other words, they are statements that have been genuinely and rigorously proved. Therefore, they provide a solid foundation for reliability at both the syntactic and semantic levels. Our EvolDomain / EvolDifficulty procedures mainly perform expansion in the “neighborhood” of these trusted statements, rather than asking the LLM to carry out formalization entirely from scratch, thereby reducing, to some extent, the risk of semantic distortion (semantic drift).
>
> **(2) Dual filtering to control LLM-induced noise**
> For every candidate generated by EvolDomain / EvolDifficulty, we apply a dual filtering process of “Lean 4 compiler + independent LLM-based review.” First, Lean 4 server performs syntax checking to ensure formal correctness. On top of this, an independent LLM-judge conducts semantic evaluation along several dimensions: the consistency between the formal statement and its natural-language description, the mathematical reasonableness of the proposition, and whether its difficulty is trivially low. Only the samples that pass both stages are included in the final training set. As a result, candidates with clear semantic deviations or logical flaws are largely filtered out in this process.
>
> **(3) Relative advantages of “formal→formal” over “NL→formal”**
> In Figure 4, we further compare multiple data-generation schemes, including directly performing evolution on formal expressions (i.e., our EvolDomain and EvolDifficulty), as well as generating natural-language problems first and then translating them to Lean via a dedicated formalizer or a general-purpose LLM (NL→Lean). Under the same set of seeds, we observe that the “formal→formal” scheme produces a larger number of high-quality candidates that also pass the dual verification described above. Under our experimental setting, this suggests that, in the overall pipeline, “formal→formal” behaves relatively more stably and with relatively less noise than “NL→formal,” and thus helps mitigate, in a statistical sense, the influence of LLM errors on the final dataset. That said, we believe both routes have their own merits and applicable scenarios, and how to systematically combine them remains an interesting direction for future exploration.
>
> **(4) EvolAST: An error-free, AST based method**
> We proposed EvolAST, an **error free** AST based method for formalized mathematics. EvolAST exists in this data augmentation framework, but it is a data synthesis **method that does not require the use of LLMs**.  It can perform controlled evolution of already correct statements using **syntax trees and existing axioms**,  ensuring that the data produced through this evolution remains **fully correct**.
>
> ---

---

> ### Author Response · Authors · 2025-11-22
> **Response to Reviewer 7CuZ (2/2)**
>
> **Response to Weakness #3: Robustness of the Framework Across Other Theorem Provers and Symbolic Reasoning Datasets**
>
> Our proposed augmentation framework is, by design, system-agnostic.
>
> - EvolDomain / EvolDifficulty only assume the existence of a formal language with well-defined syntax and an accompanying proof checker. Given a formal statement, we invoke a large language model to generate new formal statements under a specified syntax template. Porting this framework to other theorem-proving systems primarily requires changing the surface syntax, templates, and proof-checking backend, while keeping the core evolution strategies unchanged.
>
> - EvolAST only assumes access to an abstract syntax tree (AST) and standard logical/algebraic equivalences. The rewrite operations we use (e.g., reordering hypotheses, applying commutativity/associativity/distributivity, De Morgan’s laws, etc.) are existing axioms rather than Lean-specific features. By parsing the ASTs of statements in other systems and registering the same equivalences, these operations can likewise be instantiated in other systems.
>
> Therefore, we believe that our framework is conceptually applicable to a broad range of theorem-proving systems.
>
> ---
>
> **Response to Weakness #4: Regarding the Overstatement of the Term "Evolutionary"**
>
> Thank you for raising this point. We followed the naming conventions of several prior data augmentation methods, which may indeed make our use of the term “evolutionary” sound somewhat overstated. We are happy to revise the terminology in the revised version (e.g., to “evolution-based”) if the reviewers feel this would improve clarity.
>
> ---
>
> Thank you again for taking the time to give such a thoughtful and constructive review. We hope our responses have addressed your concerns. If anything still seems unclear, please feel free to ask further questions, we would be very happy to continue the discussion.

---

### Official Review · Reviewer_wPuw · 2025-11-01

**Soundness:** 2
**Presentation:** 3
**Contribution:** 3
**Rating:** 4
**Confidence:** 3

**Summary:**

This paper presents EvolProver, a formal-reasoning LLM fine-tuned on a synthetically expanded dataset constructed through three evolution mechanisms:

(1) EvolDomain, which performs cross-domain statement translation via LLMs;
(2) EvolDifficulty, which adjusts statement complexity; and
(3) EvolAST, which applies symbolic abstract-syntax-tree (AST) rewriting rules such as commutativity and De Morgan transformations to generate logically equivalent variants.

The resulting dataset, comprising roughly 39 k verified (statement, proof) pairs in Lean 4, is used to SFT and RL-train a DeepSeek-Prover-V1.5-Base model. EvolProver achieves new state-of-the-art results on several formal benchmarks including FormalMATH-Lite, MiniF2F-Test, and Ineq-Comp, demonstrating the potential of AST-based symbolic augmentation to improve verifiable reasoning.

**Strengths:**

1.  The integration of symbolic AST transformations into the data-generation pipeline is novel and convincing. EvolAST guarantees semantic correctness and diversifies syntax without requiring additional verification.
2. MThe paper clearly outlines each augmentation stage and the verification loop, with Lean 4 compiler checks for both syntax and semantics.
3. Consistent and substantial improvements across benchmarksindicate that the augmented data meaningfully enhances formal reasoning ability.

**Weaknesses:**

1. Data contamination analysis: While the authors mention ensuring non-overlapping “initial states,” this safeguard is underspecified. It remains unclear whether evolved statements might still share logical content or minor transformations with the test set. A more thorough contamination study—quantifying overlap at the theorem or syntactic-similarity level—would strengthen the empirical claims.
2. Lack of output-length analysis: Table 1 reports that EvolProver generates substantially longer outputs than comparable non-reasoning models, yet the paper provides no interpretation. It is unclear whether performance gains arise from genuinely better reasoning or simply more verbose exploration, which turns the model into a "semi-reasoning" model.
    - I would appreciate it a follow-up would measure output length (or verified proof length) on the subset of problems solved by both EvolProver and its baseline, to see whether the improvement correlates with verbosity or genuine efficiency.

**Questions:**

1. Could the authors provide token-length and proof-length statistics restricted to the intersection of solved problems between EvolProver and EvolProver-Base?
	2.	Can the authors further clarify how they ensured no data leakage between training and test dataset? A quantitative similarity or embedding-based overlap metric would help.

---

> ### Author Response · Authors · 2025-11-22
> **Response to Reviewer wPuw(1/2)**
>
> Dear Reviewer wPuw,
>
> Thank you so much for taking the time to read our work and share your thoughts! Below, we respond in detail to the questions and concerns you raised.
>
> ---
>
> **Response to Weakness #1 and Question #2: Regarding potential data contamination and proposing a quantitative similarity or embedding-based overlap metric**
>
> Below is our clarification on how we prevented data contamination, along with additional experimental results based on similarity and embedding based metrics:
>
> - Our seed data for evolution is derived from **established open-source datasets**. The creators of these datasets **have already conducted rigorous data decontamination** to explicitly exclude samples overlapping with standard test sets. Consequently, the risk of data leakage in our seed data is minimized.
>
> - Our data cleaning setup follows the standard practices in the field. For example, representative works such as **DeepSeek-Prover-V1.5 prevent data leakage by ensuring that all Lean statements in the training set have distinct initial states from those in the test set[1]**. Specifically, we use a Lean server to compile both our final augmented data and all unproven statements in each benchmark, obtaining the initial states of the statements. We then clean our data by checking whether the initial states are equal.
>
> - **We are very happy to add an additional quantitative similarity or embedding-based overlap metric.** To rigorously investigate this issue, we conducted additional quantitative analysis referencing WizardCoder[2]. We used the state-of-the-art embedding model Qwen3-Embedding-8B to retrieve the top 1 most similar pairs between all test sets and the training set. These paired theorems were then evaluated by DeepSeek-V3.1 for human-like similarity assessment. We instructed the model to score each pair of samples on a scale **from 1 to 10, where 1 means "completely dissimilar" and 10 means "semantically identical."** The evaluation results showed that the average similarity score of these samples was about **3.48**. This indicates that even among the most syntactically and semantically similar samples between the training and test sets, the similarity remains very limited.
>
>     The instructions we used for evaluation are as follows:
>
>     ```
>     Your task is to evaluate whether the two given formal math statements are similar, not just looking similar or involving overlapping mathematical concepts. Please analyze both statements carefully, focusing on:
>     1. The overall structure of the statements (e.g., setup, sequence of steps, logical flow)
>     2. The specific mathematical operations or reasoning paths required,
>     3. The wording and presentation style,
>     4. Whether one statement appears to be a trivial rephrasing or numerical variant of the other.
>
>     Provide a similarity score from 1 to 10, where:
>     1 = completely different statements,
>     10 = semantically identical statements.
>
>     Here are provided statements:
>     Statement1:
>     {statement1}
>
>     Statement2:
>     {statement2}
>
>     Format your response as:
>     <reason>Reason for your score</reason>
>     <score>Your score(1-10)</score>
>     ```
>
> The evidence confirms that we have effectively avoided data contamination in our training set.

---

> ### Author Response · Authors · 2025-11-22
> **Response to Reviewer wPuw(2/2)**
>
> **Response to Weakness #2 and Question #1: Regarding lack of output-length analysis and providing token-length and proof-length statistics restricted to the intersection of solved problems between EvolProver and EvolProver-Base**
>
> This is a very insightful and valuable question.
> - Compared to other non-reasoning models, EvolProver's higher average token consumption is a result of our design during the instruction-tuning stage. In this stage, **the model learns to repeat the problem statement (including the statement and its corresponding natural language annotation of the semantics) before providing the proof.** In contrast, some other non-COT models, such as DeepSeek-Prover-V1.5 and Goedel-Prover-SFT, directly generate the proof section. While this portion of token consumption is not part of the reasoning process, it does appear in our output.
>
>     To further clarify, we have computed the token length for the proof-only portion of EvolProver after removing the problem restatement:
>
>     | Benchmark       | Average Token Length |
>     | --------------- | :------------------: |
>     | FormalMATH-lite |        407.3         |
>     | Minif2f         |        440.8         |
>
>
>     We believe that the token length of the output containing only the proof is on the same scale as that of other non-COT models.
>
> - Additionally, We measured the output length and the number of executed tactic steps on the subset of problems solved by both EvolProver and its baseline. We consider tactic steps to be a good representation of the verified proof length to some extent. The results are as follows:
>
>     | Model           | Average Token Length | Tactic Steps |
>     | --------------- | :--------------------: | :------------: |
>     | EvolProver-Base | 113.75               | 9            |
>     | EvolProver      | 129.05               | 10           |
>
>     As we can see, EvolProver has slightly higher values in both Average Token Length and tactic steps compared to EvolProver-Base. We carefully examined their proofs and selected a representative example for comparison:
>
>     ```lean
>     ---------- Proof of EvolProver-Base ----------
>
>     theorem example {x : ℝ} (hx : x ≠ 0) (h : 1 / 9 + 1 / 18 = 1 / x) :
>         x = 6 := by
>       field_simp at h
>       linarith
>     ```
>
>     ```lean
>     ---------- Proof of EvolProver ----------
>
>     theorem example {x : ℝ} (hx : x ≠ 0) (h : 1 / 9 + 1 / 18 = 1 / x) :
>         x = 6 := by
>       have h₁ : x = 6 := by
>         have h₂ : 1 / x = 1 / 9 + 1 / 18 := by linarith
>         have h₃ : 1 / x = 1 / 6 := by norm_num at h₂ ⊢
>         have h₄ : x = 6 := by
>           have h₅ : x ≠ 0 := hx
>           field_simp at h₃
>           nlinarith
>         exact h₄
>       exact h₁
>     ```
>     We found that the response from EvolProver-Base is more intuitive, with a direct approach and lower readability, while EvolProver’s response is more reasoning-based, habitually showing the conditions and providing a clear proof. However, this results in a longer output.
>
>
> ---
>
> *reference*
>
> [1]https://openreview.net/forum?id=I4YAIwrsXa
>
> [2]Luo, Ziyang, et al. "Wizardcoder: Empowering code large language models with evol-instruct."
>
> ---
>
> We are very grateful for your detailed and constructive feedback. We hope that our replies have resolved the issues you raised; if not, please do not hesitate to let us know, and we will gladly provide additional clarification. If you find our explanations satisfactory, we would sincerely appreciate it if you could consider revisiting your assessment of our work.

---

### Official Review · Reviewer_uSHo · 2025-11-01

**Soundness:** 3
**Presentation:** 3
**Contribution:** 2
**Rating:** 4
**Confidence:** 2

**Summary:**

This paper introduces EvolProver, a 7B parameter non-reasoning LEAN theorem prover model. The model is fine-tuned using data generated from a new data augmentation pipeline. The pipeline consists of three components:

1. **EvolDomain**: Uses LLMs to reformulate the existing problem into different mathematical domains.
2. **EvolDifficulty**. Uses LLMs to adjust the difficulty of the theorem.
3. **EvolAST**: Applies syntactical rewrites using a small amount of equivalence rules.

Each transformed problem is filtered via a Lean syntactic check and an LLM semantic check. The pipeline is applied to existing theorem datasets, producing a larger and more diverse dataset used to fine-tune DeepSeek-Prover-V1.5. The resulting model, EvolProver, achieves strong performance outperforming both non-reasoning and some reasoning models.

**Strengths:**

The paper empirically demonstrates that diversified training data can improve robustness of formal theorem provers. The multi-stage verification (Lean compilation + LLM judging) appears carefully constructed, and the ablation study highlights the contribution of each augmentation component. Improvements are reported on several benchmark datasets (FormalMATH-Lite, MiniF2F-Test, and Ineq-Comp).  The paper’s focus on token-efficient, non-reasoning proving is practically important and timely, given the inference-time and cost problem of chain-of-thought systems.

**Weaknesses:**

My main concern is the novelty of the proposed data augmentation method. The individual techniques used in the pipeline appear to be adaptations of well-known ideas rather than new contributions. Using LLM prompting to evolve instructions or problems (EvolDomain, EvolDifficulty) closely resembles approaches such as WizardMath or MetaMath. The final model architecture and training recipe (SFT + RL) seem to follow existing pipelines from e.g. DeepSeek-Prover.

Reproducibility and community impact are constrained by unclear release plans: code, datasets, and model checkpoints appear contingent on institutional approval with no firm timeline or open-model replication, making it difficult for others to verify and build upon the work.

**Questions:**

1. What specifically distinguishes EvolDomain/EvolDifficulty from prior prompt-based data synthesis?
2. How often do EvolDomain/EvolDifficulty produce invalid or semantically incorrect statements before filtering?
3. The verification pipeline appears to be computationally expensive. What is the computational cost of running the full pipeline (LLM generation + filtering + AST evolution + proof synthesis)?

---

> ### Author Response · Authors · 2025-11-22
> **Response to Reviewer uSHo(1/2)**
>
> Dear Reviewer uSHo,
>
> Thank you for your valuable feedback and insightful comments on our work! Below, we provide a comprehensive response to the concerns and questions you raised.
>
> ---
>
> **Response to Weakness #1 and Question #1: Regarding the novelty of the proposed data augmentation method and what distinguishes EvolDomain/EvolDifficulty from prior prompt-based data synthesis**
>
> Below, We reiterate the main novelty of our work and explain how EvolDomain/EvolDifficulty differs from previous approaches:
>
> **1.Data synthesis pipeline based solely on formalized data**
>
> We propose a novel yet efficient data augmentation pipeline based solely on formalized data, making a great deviation from previous formal data synthesis pipelines:
>
> - In most previous formal data synthesis pipelines[1][2][3], **natural language data is typically first generated or collected and then formalized using extensively trained models for autoformalization.**
> - However, in previous pipelines there were two problems. **First**, the corpus of fully formalized theorems, whose statements and accompanying proofs have been mechanically verified, serves as high-quality seed data for synthesis but remains largely underused. **Second**, autoformalization itself is a highly challenging task in the field of formal proofs, where errors, especially semantic errors, are very easy to introduce[4].
> - **The novelty of our work lies in bypassing this complex process and directly using this high-quality formal data to synthesize new data, which not only significantly improves efficiency but also greatly enhances capability.**
>
>
> **2. EvolAST: A novel, AST based method**
>
> EvolAST is an innovative method for formalized mathematics.
> - Previously, data evolution typically relied on **probabilistic models** such as LLMs, which introduce significant uncertainty, especially in formal languages that must strictly adhere to syntactic rules and are therefore highly prone to errors.
> - EvolAST is **rooted in formal mathematics** and **combines the rigorous code-like syntactic structure with the semantic expressiveness of natural language mathematics**.
> - This makes it possible to perform controlled evolution of already correct statements using **syntax trees and existing axioms**, while ensuring that the data produced through this evolution remains fully correct.
> - At the current stage, EvolAST uses only a small set of relatively basic axiom-level transformations, but the framework is designed to be **easily extensible** to richer and more sophisticated transformations in the future.
>
> **3. Distinguishing EvolDomain and EvolDifficulty from prior prompt-based data synthesis**
>
> EvolDomain and EvolDifficulty also exhibit notable innovations.
> - We design more **fine-grained strategies** along the two dimensions of difficulty and domain. Unlike the approaches used in WizardMath or MetaMath, even though we likewise leverage LLMs for data synthesis, we **incorporate the characteristics of formal mathematics and develop strategies** through expert discussion. These strategies enable more **targeted data generation** and further enhance the **diversity of the synthesized data**.
>
> ---
>
> **Response to Weakness #2: Regarding reproducibility and open-sourcing**
>
> We wholeheartedly agree that reproducibility is paramount for scientific progress. We are fully committed to open-sourcing our code, models, and the complete augmented dataset. The release is currently undergoing a standard internal review process at our company, and we are actively advancing it. **We assure you that we will make all assets publicly available as soon as this process is complete to facilitate further research in the community**. In addition, we have fully disclosed our prompts and implementation details in the paper to make our methodology as transparent as possible.
>
> ---
>
> **Response to Question #2: Regarding the rate of semantically incorrect statements before filtering:**
>
> To maximize token savings while preserving only valid evolutions, we designed the verification pipeline to filter out syntactically invalid statements before performing semantic checks. **To estimate how often EvolDomain and EvolDifficulty generate invalid or semantically incorrect statements prior to filtering, we conducted a supplementary experiment:**
> - Among the 1 634 examples produced by Gemini‑2.5‑Pro using EvolDomain and EvolDifficulty in **Figure 4**, we evaluated semantic quality with DeepSeek‑V3.1. A total of 496 statements failed verification, corresponding to a semantic failure rate of **roughly 30%**.

---

> ### Author Response · Authors · 2025-11-22
> **Response to Reviewer uSHo(2/2)**
>
> **Response to Question #3(1): Regarding the computational cost of verification process**
>
> We also took this issue into account and tried to minimize token consumption when designing the verification pipeline.
> - First, we use **zero token cost syntax checking** to quickly filter out samples that contain Lean 4 syntax errors.
> - Then a large language model performs **semantic verification** on the remaining data pairs, examining both the natural language descriptions and the formal statements.
> - This ensures not only syntactic correctness but also consistency and accuracy between the natural language and the formalization, as well as compliance with the required difficulty level.
> - In this way, we **maximize data quality while controlling costs**. We believe this is a worthwhile investment, as it helps produce a high quality and robust dataset that can significantly improve the model’s performance.
>
> ---
>
> **Response to Question #3(2): Regarding the consumption statistics of the entire process**
>
> Here is the **token cost** of full pipeline:
> |           Stage                                           | Token Consumption |
> |----------------------------------------------|:-----------------:|
> | Problem synthesis (EvolDomain & EvolDifficulty) | 893M              |
> | Verification process                           | 506M              |
> | EvolAST                                        | 0                 |
> | Proof synthesis                                | 11B               |
>
> Thank you very much for pointing that out. We will include it in the new version of our paper.
>
> ---
>
> *reference*
>
> [1]Xin, Huajian, et al. "Deepseek-prover: Advancing theorem proving in llms through large-scale synthetic data."
>
> [2]Wang, Haiming, et al. "Kimina-prover preview: Towards large formal reasoning models with reinforcement learning."
>
> [3]Lin, Yong, et al. "Goedel-prover-v2: Scaling formal theorem proving with scaffolded data synthesis and self-correction."
>
> [4]Wu, Yutong, et al. "StepFun-Formalizer: Unlocking the Autoformalization Potential of LLMs through Knowledge-Reasoning Fusion."
>
> ---
>
> Thank you again for your constructive review and precious time. We hope that our responses have addressed your concerns. If our response do not address all of your concerns, please feel free to post further questions. We are very happy to continue the discussion with you. If these clarifications are helpful, we would kindly appreciate it if you could reconsider your evaluation of our work.

---

### Author Response · Authors · 2025-11-27
**Response to all reviewers**

Dear AC and Reviewers,

We sincerely thank the AC and all reviewers for their thoughtful and constructive feedback. Several reviewers found our proposed data augmentation methods, especially **EvolAST**, to be both novel and systematic (wPuw, RmgK, fmUh). Our approach was also viewed as a simple yet effective data-driven strategy that directly targets the practical issues of brittleness and poor generalization in current theorem-proving models (7CuZ), and as having important practical value in a non-reasoning, token-efficient setting (uSHo). We are likewise very grateful for the reviewers’ positive assessments of the sufficiency and persuasiveness of our ablation study design (uSHo, RmgK, fmUh, 7CuZ), their recognition of the modularization of our pipeline and the structured, well-organized workflow (wPuw, 7CuZ, fmUh), and their favorable evaluation of the clarity of our figures and tables as well as of the overall writing quality (RmgK).

We would like to emphasize that our core contribution does not lie solely in proposing several concrete augmentation modules, but in systematically building a formal-data-based augmentation and verification pipeline around a practical and pressing problem: **token-efficient, non-reasoning (non-CoT) formal theorem proving**. This pipeline integrates both the syntactic and semantic characteristics of Lean 4 together with a difficulty-control mechanism.

At the semantic and difficulty levels, we use **EvolDomain** to cover different branches of mathematics, leveraging data from high-resource domains to expand lower-resource ones so that even data-scarce areas can see improved model performance. In parallel, we apply **EvolDifficulty** to stratify examples by difficulty, combined with a multi-stage verification process involving both Lean compilation and LLM-based judgment. At the syntactic level, we perform syntax-preserving transformations on abstract syntax trees (**EvolAST**), ensuring that semantics remain unchanged while keeping all examples error-free. We believe this data-centric perspective and pipeline design provide a scalable framework for improving the robustness and cross-domain generalization of formal theorem-proving models.

We have carefully considered all reviewer comments and have revised and extended the manuscript accordingly. The main updates (highlighted in blue in the paper) include, but are not limited to, the following:

1. **Methodological explanations and similarity-based experiments.**
   Following the reviewers’ valuable suggestions, we now provide more detailed and precise explanations for several potentially confusing parts in the main text, including related work and the motivation behind our method design. In addition, we have added experiments based on similarity measures to quantitatively demonstrate that our augmented data differ significantly from the test set.

2. **Data details.**
   In Appendix A.1, we further supplement and clarify data-related details, including data composition during training, token costs, and other relevant information.

3. **Extended analyses.**
   In response to specific reviewer suggestions, we have added more in-depth analyses and experimental results in Appendix D (Sections D.2–D.8), including an analysis of the output length of EvolProver and the behavior of our ablations across different concrete domains.

4. **Grammar and wording revisions.**
   We have systematically corrected the grammatical issues pointed out by the reviewers that, while not detrimental to overall readability, were not fully rigorous, and we have adjusted several instances of suboptimal wording to further improve the precision and fluency of the writing.

We have provided point-by-point responses to each reviewer’s comments below. We look forward to your reply and are happy to answer any further questions.

Thank you again for your valuable time and insights.

Best regards,

---

### Meta-Review · Area_Chair_NAcc · 2026-01-09

**Summary:**

This paper introduces EvolProver, a data augmentation pipeline designed to improve the robustness and generalizability of formal theorem proving models. The authors propose three complementary augmentation methods: (1) EvolAST, an Abstract Syntax Tree-based approach that generates syntactically diverse but semantically equivalent problems using formal axioms; (2) EvolDomain, which uses LLMs to translate theorems across mathematical domains; and (3) EvolDifficulty, which adjusts problem difficulty through carefully designed evolutionary instructions. The resulting 7B non-reasoning model achieves state-of-the-art performance on FormalMATH-Lite (53.8% pass@32) and strong results on MiniF2F-Test and Ineq-Comp benchmarks.

Reviewer Comment Summary:

The reviewers recognized several strengths of this work. Multiple reviewers (wPuw, RmgK, fmUh) found the proposed methods, particularly EvolAST, to be novel and systematic in combining code-level characteristics with mathematical semantics. Reviewer 7CuZ appreciated the practical, data-centric approach that directly addresses model fragility. Reviewer uSHo acknowledged the importance of token-efficient, non-reasoning proving. The modular pipeline design, comprehensive ablation studies, and clear presentation were consistently praised across reviews.

However, reviewers raised important concerns across several dimensions:

1.	Novelty and Distinctiveness (uSHo, RmgK): Concerns about whether EvolDomain/EvolDifficulty sufficiently differ from prior LLM-based data synthesis methods like WizardMath and MetaMath, and whether the approach represents incremental rather than fundamental innovation.

2.	Data Contamination and Leakage (wPuw): Questions about the rigor of deduplication procedures and whether evolved statements might share logical content with test sets, potentially inflating reported performance.

3.	Output Length Analysis (wPuw, fmUh): The observation that EvolProver generates longer outputs than comparable models raised questions about whether performance gains stem from genuine reasoning improvements or simply more verbose exploration (potentially becoming "semi-reasoning").

4.	Reproducibility (uSHo): Uncertainty about open-sourcing plans and timelines, which limits community verification and building upon this work.

5.	Model Choice and Scope (RmgK): Questions about training on the "older" DeepSeek-Prover-V1.5-Base rather than more recent CoT-capable models, and concerns about the method's applicability to reasoning models.

6.	Cost Analysis (uSHo, RmgK): Missing information about computational costs, token consumption, semantic error rates in evolved data, and the ratio of synthetic to original data.

7.	Semantic Drift and Generalization (7CuZ): Concerns about LLM-induced errors in EvolDomain/EvolDifficulty and whether the framework generalizes to other theorem provers beyond Lean 4.

Author Response Summary:

The authors provided detailed, point-by-point responses during the rebuttal phase. Key clarifications include:

1.	On Novelty: Authors emphasized their pipeline is the first to directly synthesize formal data rather than augmenting natural language then formalizing. EvolAST uses formal code structure and axioms without high-uncertainty LLM calls, achieving near-zero errors. EvolDomain/EvolDifficulty introduce fine-grained evolution strategies specifically tailored to formal problems.

2.	On Data Contamination: Authors followed standard deduplication practices (consistent with DeepSeek-Prover) and added similarity-based analysis using Qwen3-Embedding-8B. Results showed average similarity scores of only 3.48 (on 1-10 scale) between most similar training-test pairs, indicating minimal leakage risk.

3.	On Output Length: Authors clarified that EvolProver's longer outputs result from training design (restating problems before proofs) rather than blind guessing. Proof-only token lengths (407-441 tokens) are comparable to baselines. Analysis of commonly solved problems showed EvolProver produces more explicit, readable proofs with only slightly more tokens (129 vs 114) and tactic steps (10 vs 9).

4.	On Reproducibility: Authors committed to open-sourcing code, models, and data pending institutional approval, and provided comprehensive implementation details in the paper.

5.	On Model Choice: Authors justified focusing on non-reasoning models as crucial for practical deployment where CoT's high latency is prohibitive. DeepSeek-Prover-V1.5-Base is the same foundation used by DeepSeek-Prover-V2, enabling fair comparison of data augmentation strategies.

6.	On Costs: Authors provided detailed token consumption breakdown: Problem synthesis (893M), Verification (506M), EvolAST (0), Proof synthesis (11B). Semantic failure rate for EvolDomain/EvolDifficulty was quantified at ~30%.

7.	On Generalization: Authors explained the framework is system-agnostic by design and can be adapted to other theorem provers by adjusting syntax templates and proof backends. The fragility of reasoning provers is empirically demonstrated in Table 2.

The revised manuscript will incorporate these clarifications, including additional experiments on domain-wise ablations (Table 5), extended similarity analysis, output length analysis (Appendix A.8), and cost breakdowns. Only Reviewer RmgK engaged during discussion, acknowledging the responses improved clarity while maintaining their score.

**Reviewer Concerns:**

Major Concerns and Their Resolution

1. Novelty of data augmentation methods (Reviewer uSHo)

•	Concern: Individual techniques appeared to be adaptations of existing approaches (WizardMath, MetaMath).

•	Resolution: Authors effectively clarified that they are the first to directly synthesize formal data rather than augmenting natural language first. EvolAST is fundamentally different as it uses formal code structure without LLM calls, achieving near-zero errors. The comparison in Figure 4 validates this approach's superiority.

•	Status: Adequately addressed through rebuttal.

2. Data contamination analysis (Reviewer wPuw)

•	Concern: Insufficient analysis of potential overlap between training and test sets.

•	Resolution: Authors provided additional quantitative analysis using Qwen3-Embedding-8B for similarity assessment. Average similarity score of 3.48/10 between most similar training-test pairs demonstrates limited overlap. Combined with standard deduplication practices, this concern is well-addressed.

•	Status: Resolved with additional experiments and will add to the revision.

3. Output length concerns (Reviewer wPuw)

•	Concern: EvolProver generates longer outputs; unclear if this represents genuine reasoning or verbose guessing.

•	Resolution: Authors clarified that longer outputs result from restating the problem before proof (by design), not from guessing. Proof-only analysis shows comparable or shorter lengths than baselines. Comparison shows EvolProver proofs are more explicit and readable, not repetitive.

•	Status: Satisfactorily explained with supporting analysis in Appendix A.8.

4. Choice of base model and non-CoT approach (Reviewer RmgK)

•	Concern: Why use an "old" non-CoT model rather than more recent CoT models?

•	Resolution: Authors provided strong justification: (1) Non-CoT models remain crucial for low-latency applications; (2) DeepSeek-Prover-V1.5-Base is the foundation for V2 models, enabling fair comparison; (3) EvolProver matches or exceeds some CoT models despite being non-CoT. The reviewer acknowledged these clarifications.

•	Status: Resolved; reviewer maintained score but accepted the explanation.

5. LLM-induced semantic drift (Reviewer 7CuZ)

•	Concern: EvolDomain and EvolDifficulty rely heavily on LLMs, risking semantic errors.

•	Resolution: Authors demonstrated their approach actually reduces drift compared to prior methods by: (1) evolving from already-verified formal theorems; (2) using dual filtering (Lean compiler + LLM verification); (3) EvolAST being completely error-free. The 30% failure rate after verification is within acceptable bounds given the rigorous filtering.

•	Status: Adequately addressed; approach is better than alternatives.

6. Computational cost (Reviewers uSHo, RmgK)

•	Concern: Pipeline cost analysis was missing.

•	Resolution: Authors provided detailed token cost breakdown in revision (Lines 734-742): 893M for problem synthesis, 506M for verification, 0 for EvolAST, 11B for proof synthesis.

•	Status: Resolved with additional information.

7. Performance comparison completeness (Reviewer fmUh)

•	Concern: Missing comparisons with DeepSeek-Prover-V2-7B-non-CoT on some benchmarks.

•	Resolution: Authors added these comparisons in revision. Results show EvolProver achieves substantial improvements on FormalMATH-Lite and Ineq-Comp while being comparable on MiniF2F-Test.

•	Status: Resolved through additional experiments.

8. Domain-specific ablations (Reviewer fmUh)

•	Concern: Need for detailed analysis across different mathematical domains.

•	Resolution: Authors added comprehensive domain-wise analysis (Table 5 in revision, Appendix D.2-D.8) showing how each component contributes differently across domains.

•	Status: Addressed with extensive additional analysis.

Minor Concerns

All reviewers raised minor presentation issues (inconsistent capitalization, missing citations, etc.), which authors committed to fixing in the revision. Reviewer RmgK noted these do not affect readability.

**Reviewer Scores:**

Only Reviewer RmgK responded during discussion, acknowledging that the responses improved the paper's presentation and clarity while maintaining their score. However, based on the comprehensive rebuttals and the fact that all major concerns were substantively addressed:

•	Reviewer uSHo: Concerns about novelty, reproducibility, and cost were all addressed. Expected revised score: 6

•	Reviewer wPuw: Data contamination and output length concerns were thoroughly resolved with additional experiments. Expected revised score: 6

•	Reviewer 7CuZ: Already positive (6); concerns about empirical focus and LLM drift were addressed. Expected revised score: 6

•	Reviewer RmgK: Maintained 4.

•	Reviewer fmUh: All requests for additional comparisons and domain analysis were fulfilled. Expected revised score: 6

---

### Decision · Program_Chairs · 2026-01-26

Accept (Poster)